



# Rainfall estimation from a German-wide commercial microwave link network: Optimized processing and validation for one year of data

Graf Maximilian[1], Chwala Christian[1,2], Polz Julius[1], and Kunstmann Harald[1,2]

[1]Karlsruhe Institute of Technology, IMK-IFU, Kreuzeckbahnstr. 19, 82467 Garmisch-Partenkirchen, Germany
[2]University Augsburg, Institute for Geography, Alter Postweg 118, 86159 Augsburg, Germany

**Correspondence:** Graf Maximilian (maximilian.graf@kit.edu)

**Abstract.** Rainfall is one of the most important environmental variables. However, it is a challenge to measure it accurately over space and time. During the last decade commercial microwave links (CMLs) operated by mobile network providers have proven to be an additional source of rainfall information to complement traditional rainfall measurements. In this study we present the processing and evaluation of a German-wide data set of CMLs. This data set was acquired from around 4000 CMLs distributed across Germany with a temporal resolution of one minute. The analyzed period of one year spans from September 2017 to August 2018. We compare and adjust existing processing schemes on this large CML data set. For the crucial step of detecting rain events in the raw attenuation time series, we are able to reduce the amount of miss-classification. This was achieved by a new approach to determine the threshold which separates a rolling window standard deviation of the CMLs signal into wet and dry periods. For the compensation of wet antenna attenuation, we compare a time-dependent model with a rain-rate-dependent model and show that the rain-rate-dependent method performs better for our data. As precipitation reference, we use RADOLAN-RW, a gridded gauge-adjusted hourly radar product of the German Meteorological Service (DWD), from which we derive the path-averaged rain rates along each CML path. Our data processing is able to handle CML data across different landscapes and seasons very well. For hourly, monthly and seasonal rainfall sums we found high agreement between CML-derived rainfall and the reference, except for the cold season with non-liquid precipitation. This analysis shows that opportunistic sensing with CMLs yields rainfall information with a quality similar to gauge-adjusted radar data during periods without non-liquid precipitation.





# 1 Introduction

Measuring precipitation accurately over space and time is challenging due to its high spatiotemporal variability. It is a crucial component of the water cycle and knowledge of the spatiotemporal distribution of precipitation is an important quantity in many applications across meteorology, hydrology, agriculture, and climate research.

Typically, precipitation is measured by rain gauges, ground-based weather radars or spaceborne microwave sensors. Rain gauges measure precipitation at the point scale. Errors can be caused for example by wind, solid precipitation or evaporation losses (Sevruk, 2005). The main disadvantage of rain gauges is their lack of spatial representativeness.

Weather radars overcome this spatial constraint, but are affected by other error sources. They do not directly measure rainfall but estimate it from related observed quantities, typically via the Z-R relation which links the radar reflectivity "Z" to the
rain rate "R". This relation, however, depends on the rain drop size distribution (DSD), resulting in significant uncertainties. Dual-polarization weather radars reduce this uncertainties, but still struggle with the DSD-dependence of the rain rate estimation (Berne and Krajewski, 2013) . Additional error sources can stem from the measurement high above ground, from beam blockage or ground clutter effects.

Satellites can observe large parts of the earth, but their spatiotemporal coverage is restricted by their orbits. Typical revisit
15  times are in the order of hours to days. As a result, even merged multi-satellite products have a latency of several hours, e.g. the Integrated Multi-satellite Retrievals (IMERG) early run of the Global Precipitation Measurement Mission (GPM) has a latency of 6 hours. The employed retrieval algorithms are highly sophisticated and several calibration and correction stages are potential error sources (Maggioni et al., 2016).

Additional rainfall information, for example derived from commercial microwave links (CMLs) maintained by cellular net-
work providers, can be used to compare and complement existing rainfall data sets (Messer et al., 2006). In regions with sparse observation networks, they might even provide unique rainfall information.

The idea to derive rainfall estimates via the opportunistic usage of attenuation data from CML networks emerged over a decade ago independently in Israel (Messer et al., 2006) and the Netherlands (Leijnse et al., 2007). The main research foci in the first decade of dedicated CML research were the development of processing schemes for the rainfall retrieval and the reconstruction
of rainfall fields. The first challenge for rainfall estimation from CML data is to distinguish between fluctuations of the raw attenuation data during rainy and dry periods. This was addressed by different approaches which either compared neighbouring CMLs using the spatial correlation of rainfall (Overeem et al., 2016a) or which focused on analyzing the time series of individual CMLs (Chwala et al., 2012; Schleiss and Berne, 2010; Wang et al., 2012). Another challenge is to estimate and correct the effect of wet antenna attenuation. This effect stems from the attenuation caused by water droplets on the covers of CML
antennas, which leads to rainfall overestimation (Fencl et al., 2019; Leijnse et al., 2008; Schleiss et al., 2013).

Since many hydrological applications require spatial rainfall information, several approaches have been developed for the generation of rainfall maps from the path-integrated CML measurements. Kriging was successfully applied to produce countrywide rainfall maps for the Netherlands (Overeem et al., 2016b), representing CML rainfall estimates as synthetic point observation at the center of each CML path. More sophisticated methods can account for the path-integrated nature of the CML observations,





using an iterative inverse distance weighting approach (Goldshtein et al., 2009), stochastic reconstruction (Haese et al., 2017) or tomographic algorithms (D'Amico et al., 2016; Zinevich et al., 2010).

CML-derived rainfall products were also used to derive combined rainfall products from various sources (Fencl et al., 2017; Liberman et al., 2014; Trömel et al., 2014). In parallel, first hydrological applications were tested. CML-derived rainfall was

used as model input for hydrologic modelling studies for urban drainage modeling with synthetic (Fencl et al., 2013) and real world data (Stransky et al., 2018) or on run-off modeling in natural catchments (Brauer et al., 2016; Smiatek et al., 2017).

With the exception of the research carried out in the Netherlands, where more than two years of data from a country-wide CML network were analyzed (Overeem et al., 2016b), CML processing methods have only been tested on small data sets. We advance the state of the art by performing an analysis of rainfall estimates derived from a German-wide network of close to

4000 CMLs. In this study one CML is counted as the link along one path with typically two sub-links, for the communication in both directions. The temporal resolution of the data set is one minute and the analyzed period is one year from September 2017 until August 2018. The network covers various landscapes from the North German Plain to the Alps in the south which feature individual precipitation regimes.

The objectives of this study are (1) to compare and adjust selected existing CML data processing schemes for the classification

of wet and dry periods and for the compensation of wet antenna attenuation and (2) to validate the derived rainfall rates with an established rainfall product, namely RADOLAN-RW, both on the country-wide scale of Germany.

## 2   Data

### 2.1   Reference data set

The *Radar-Online-Aneichung* data set (RADOLAN-RW) of the German Weather Service (DWD) is a radar-based and gauge

adjusted precipitation data set. We use data from the archived real-time product RADOLAN-RW as reference data set throughout this work (DWD). It is compiled from 16 weather radars operated by DWD and adjusted by 1100 rain gauges in Germany and 200 rain gauges from surrounding countries. The gridded data set has a spatial resolution of 1 km covering Germany with 900 by 900 grid cells. The temporal resolution is one hour and the minimal detection limit of rainfall is 0.1 mm (Bartels et al., 2004; Winterrath et al., 2012).

Kneis and Heistermann (2009) and Meissner et al. (2012) compared RADOLAN-RW products to gauge-based data sets for small catchments and found differences in daily, area averaged precipitation sums of up to 50 percent, especially for the winter season. Nevertheless, no data set with comparable temporal and spatial resolution, as well as extensive quality control is available.

In order to compare the path integrated rainfall estimates from CMLs and the gridded RADOLAN-RW product, RADOLAN-

RW rainfall rates are resampled along the individual CML paths. For each CML the weighted average of all intersecting RADOLAN-RW grid cells is calculated, with the weights being the lengths of the intersecting CML path in each cell. As result, one time series of the hourly rain rate is generated from RADOLAN-RW for each CML.





## 2.2 Commercial Microwave Link Data

We present data of 3904 CMLs operated by Ericsson in Germany. Their distribution over Germany is shown in Fig. 1. The CMLs are distributed country-wide over all landscapes in Germany, ranging from the North German Plain to the Alps in the south. The uneven distribution, with large gaps in the north east can be explained by the fact that we only access one subset of

all installed CMLs, only Ericsson MINI-LINK Traffic Node systems operated for one cell phone provider.

CML data is retrieved with a real-time data acquisition system which we operated in cooperation with Ericsson (Chwala et al., 2016). Every minute, the current transmitted signal level (TSL) and received signal level (RSL) are requested from more than 4000 CMLs for both ends of each CML. The data is then immediately sent to and stored at our server. For the analysis presented in this work, we use this 1-minute instantaneous data of TSL and RSL for the period from September 2017 to August 2018 for

3904 CMLs. Due to missing, unclear or corrupted metadata we cannot use all CMLs. Furthermore, we only use data of one sub-link per CML, i.e. we only use one pair of TSL and RSL out of the two that are available for each CML.

The available power resolution is 1 dB for TSL and 0.3 (with occasional jumps of 0.4 dB) for RSL. While the length of the CMLs ranges between a few hundred meters to over 30 km most CMLs have a length of 5 to 10 km. They are operated with frequencies ranging from 10 to 40 GHz, depending on their length. Figure 2 shows the distributions of path lengths and

frequencies. For shorter CMLs higher frequencies are used.

To derive rainfall from CMLs, we use the difference between TSL and RSL, the transmitted minus received signal level (TRSL). An example of a TRSL time series is shown in Fig. 3a). To compare the rain rate derived from CMLs with the reference rain rate, we resample it from a minutely to an hourly resolution after the processing.

In our data set 2.2 percent are missing time steps due to outages of the data acquisition systems. Additionally 1.2 percent of the

raw data show missing values (Nan) and 0.1 percent show default fill values (e.g. -99.9 or 255.0) of the CML hardware, which we exclude from the analysis. Furthermore we have to remove 9.9 percent of data, because of inconsistent TSL records for CMLs with a so called 1+1 hot standby system, i.e. which have a second backup radio unit installed, which shares one antenna with the main unit.

The size of this CML data set is approximately 100 GB in memory. The data set is operationally continuously extended by the

data acquisition allowing also the possibility of near-realtime rainfall estimation.

## 3 Methods

### 3.1 Performance measures

To evaluate the performance of the CML-derived rain rates against the reference data set, we used several measures which we calculated on an hourly basis. We defined a confusion matrix according to Tab. 1 where *wet* and *dry* refer to hours with and

without rain, respectively. Hours with less than 0.1 mm/h were considered as dry in both data sets. The Matthew's correlation coefficient (MCC) summarizes the four values of the confusion matrix in a single measure (1) and is typically used as measure of binary classification in machine learning. This measure is accounting for the skewed ratio of wet and dry events. It is high





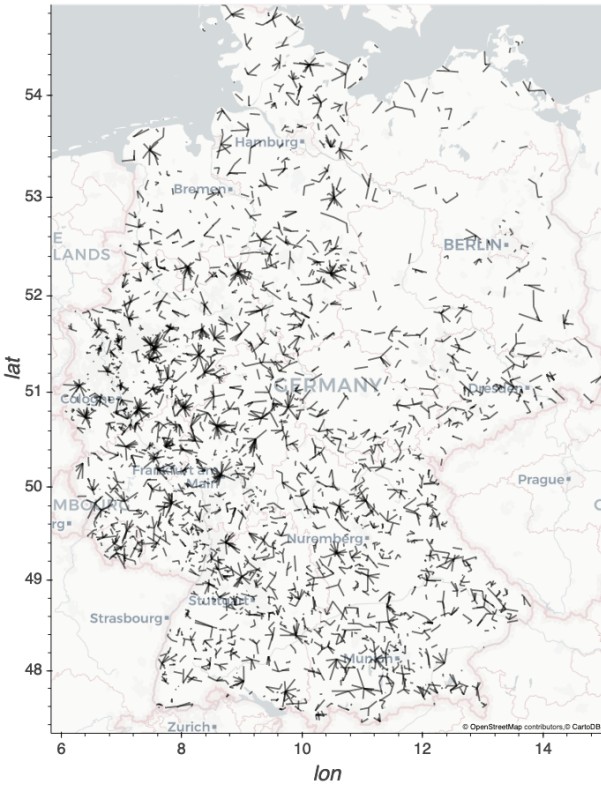

**Figure 1.** Map of the distribution of 3904 CMLs over Germany. © OpenStreetMap contributors 2019. Distributed under a Creative Commons BY-SA License.

**Table 1.** Adopted confusion matrix

|  |  | reference | |
|---|---|---|---|
|  |  | *wet* | *dry* |
| CML | *wet* | true wet (TP) | false wet (FP) |
|  | *dry* | missed wet (FN) | true dry (TN) |

only if the classifier is performing well on both classes.

$$\text{MCC} = \frac{\text{TP} * \text{TN} - \text{FP} * \text{FN}}{\sqrt{(\text{TP} + \text{FP})(\text{TP} + \text{FN})(\text{TN} + \text{FP})(\text{TN} + \text{FN})}} \tag{1}$$

The mean detection error (MDE) (2) is introduced as a further binary measure focusing on the miss-classification of rain events.

$$\text{MDE} = \frac{\frac{\text{FN}}{\text{n(wet)}} + \frac{\text{FP}}{\text{n(dry)}}}{2} \tag{2}$$





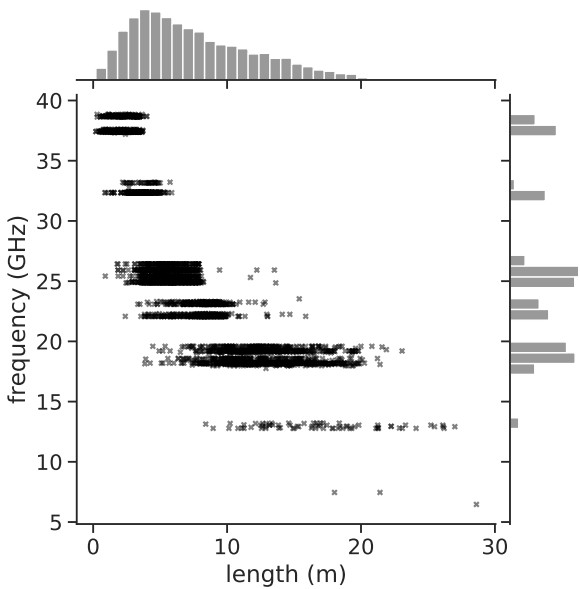

**Figure 2.** Scatterplot of the length against the microwave frequency of 3904 CMLs including the distribution of length and frequency.

It is calculated as the average of missed wet and false wet rates of the contingency table from Tab. 1.

The linear correlation between CML-derived rainfall and the reference is expressed by the Pearson correlation coefficient (PCC). The coefficient of variation (CV) in (3) gives the distribution of CML rainfall around the reference expressed by the ratio of residual standard deviation and mean reference rainfall,

$$CV = \frac{\mathrm{std}\sum(R_{CML} - R_{reference})}{R_{reference}} \tag{3}$$

where $R_{CML}$ and $R_{reference}$ are hourly rain rates of the respective data set. Furthermore we computed the mean absolute error (MAE) and the root mean squared error (RMSE) to measure the accuracy of the CML rainfall estimates.

### 3.2 From raw signal to rain rate

As CMLs are an opportunistic sensing system rather than part of a dedicated measurement system, data processing has to be done with care. Most of the CML research groups developed their own methods tailored to their needs and data sets. Overviews of these methods are summarized by Chwala and Kunstmann (2019) and Uijlenhoet et al. (2018).

The size of our data set is a challenge itself. As TRSL can be attenuated by rain or other sources, described in 3.2.1 and only raw RSL and RSL data is provided, the large size of the data set is of advantage but also a challenge. Developing and evaluating methods requires repeatedly testing with the complete data set. This requires an automated processing workflow, which we implemented as a parallelized workflow on a HPC system using the Python packages *xarray* and *dask* for data processing and





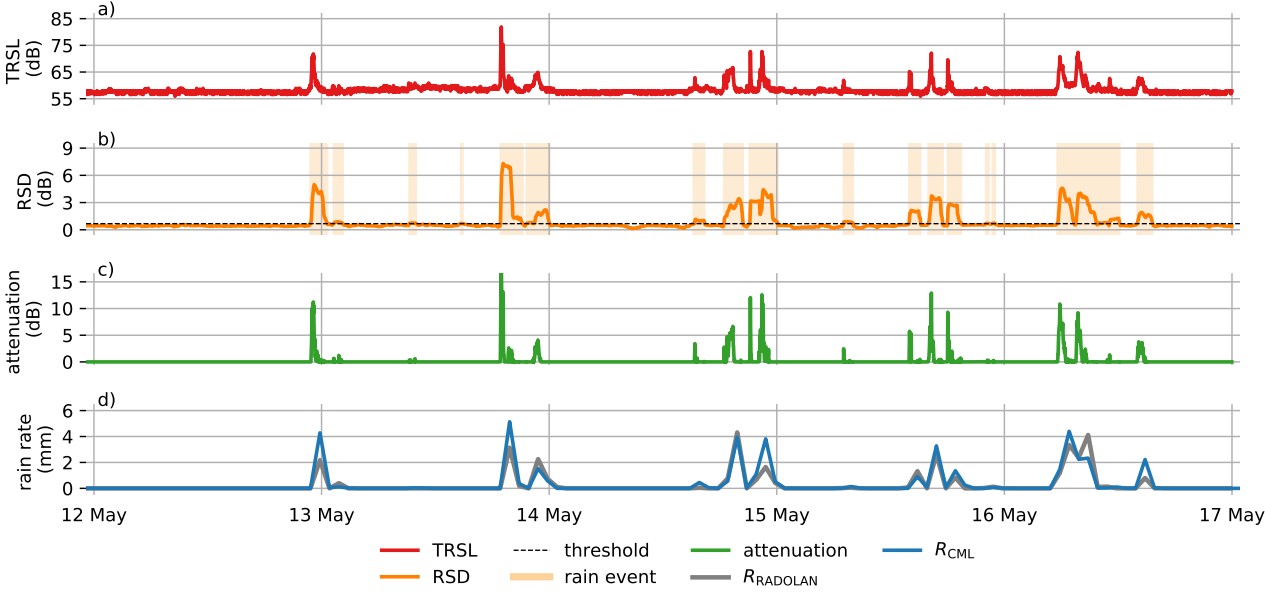

**Figure 3.** Processing steps from the TRSL to rain rate. a) TRSL is the difference of TSL - RSL, the raw transmitted and received signal level of a CML b) RSD (rolling standard deviation) of the TRSL with an exemplary threshold and resulting wet and dry periods, c) Attenuation is the difference between the baseline and the TRSL during wet periods and d) derived rain rate resampled to an hourly scale in order to compare it to the reference RADOLAN-RW

visual exploration. The major challenges which arised from the processing of raw TRSL data into rain rates and the selected methods from literature are described in the following sections.

### 3.2.1 Erratic behavior

Rainfall is not the only source of attenuation of microwave radio along a CML path. Additional attenuation can be caused by
5   atmospheric constituents like water vapor or oxygen, but also by refraction, reflection or multi-path propagation of the beam (Upton et al., 2005). In particular, refraction, reflection and multi-path propagation can lead to strong attenuation in the same magnitude as from rain. CMLs that exhibit such behavior have to be omitted due to their noisiness.

We excluded erratic CML data which was extremely noisy, showed drifts or jumps, from our analysis. We omitted individual CMLs in a sanity check when 1) a five hour moving window standard deviation exceeds the threshold 2 for more then ten
10  percent of a month, or 2) a one hour moving window standard deviation exceeds the threshold 0.8 more than 33 percent of a month. This removes data that shows unreasonably high amount of strong fluctuations.

Jumps in data are mainly caused by single default values in the TSL which are described in 2.2. When we removed these default values, we are able to remove the jumps. TRSL can drift and fluctuate on daily and yearly scale (Chwala and Kunstmann, 2019). We could neglect the influence of these drifts in our analysis, because we dynamically derived a baseline for each rain event, as





explained in section 3.2.2. We also excluded CMLs having a constant TRSL over a whole month. Overall, we have excluded 405 CMLs completely from our country-wide analysis.

### 3.2.2 Rain event detection and baseline estimation

The TRSL during dry periods can fluctuate over time due to ambient conditions as mentioned in the previous section. Rainfall
produces additional attenuation on top of the dry fluctuation. In order to calculate the attenuation from rainfall, a baseline level of TRSL during each rain event has to be determined. We derived the baseline from the precedent dry period. During the rain event, this baseline was held constant, as no additional information on the evolution of the baseline level is available. The crucial step for deriving the baseline is to separate the TRSL time series into wet and dry periods, because only then the correct reference level before a rain event is used. By subtracting the baseline from TRSL, we derived the attenuation caused
by rainfall which is shown in Fig. 3c).

The separation of wet and dry periods is essential, because the errors made in this step will impact the performance of rainfall estimation. Missing rain events will result in rainfall underestimation. False detection of rain events will lead to overestimation. The task of detecting rain events in the TRSL time series is simple for strong rain events, but challenging when the attenuation from rain is approaching the same order of magnitude as the fluctuation of TRSL data during dry conditions.

There are two essential concepts to detect rain events. One compares the TRSL of a certain CML to neighbouring CMLs (Overeem et al., 2016a) and the other investigates the time series of each CML separately (Chwala et al., 2012; Schleiss and Berne, 2010; Wang et al., 2012). We choose the latter one and use a rolling standard deviation (RSD) with a centered moving window of 60 minutes length as a measure for the fluctuation of TRSL as proposed by Schleiss and Berne (2010).

It is assumed that RSD is high during wet periods and low during dry periods. Therefore, an adequate threshold must be
defined, which differentiates the RSD time series in wet and dry periods. An example of an RSD time series and a threshold is shown in Fig. 3b) where all data points with RSD values above the threshold are considered as wet.

Schleiss and Berne (2010) proposed the use of a RSD threshold derived from rain fall climatology e.g. from nearby rain gauges. For our data set we assume that it is raining 5 percent of all minutes in Germany. Therefore, we use the 95 percent quantile of RSD as a threshold, assuming that the 5 percent of highest fluctuation of the TRSL time series refer to the 5 percent of rainy
periods.

We call this threshold the climatologic threshold and compare it to two new definitions of thresholds. For the first new definition we derive the optimal thresholds for each CML based on our reference data for the month of May 2018. The MCC between each CML and its reference is optimized to get the best threshold for each CML in this month. Each CMLs threshold from this month is then used for the whole analysis period.

The second new definition to derive a threshold is based on the quantiles of the RSD, similarly to the initially proposed method by Schleiss and Berne (2010). However, we propose to not focus on the fraction of rainy periods for finding the optimal threshold, since a rainfall climatology is likely not valid for individual years and not easily transferable to different locations. We take the 80th quantile as a measure of the strength of the TRSL fluctuation during dry periods for of each CML and multiply it by a factor to derive an individual threshold. The 80th quantile is different to the climatologic threshold, as this





quantile represents the general notion of each TRSL time series to fluctuate rather than the percentage of time in which it is raining. We chose the 80th quantile, since it is very unlikely that it is raining 20 percent of the time in a month or more in Germany.

To find the right factor we selected the month of May 2018 and fitted a linear regression between the optimal threshold for
each CML and the 80th quantile. The optimal threshold was derived beforehand with a MCC optimization from the reference. We used this factor throughout the year as we found it to be similar for all months of the analyzed period.

### 3.2.3   Wet antenna attenuation

Wet antenna attenuation is the attenuation caused by water on the cover of a CML antenna. With this additional attenuation, the derived rain rate overestimates the true rain rate (Schleiss et al., 2013; Zinevich et al., 2010). The estimation of WAA is
complex, as it is influenced by partially unknown factors, e.g. the material of the antenna cover. van Leth et al. (2018) found differences in WAA magnitude and temporal dynamics due to different sizes and shapes of the water droplets on hydrophobic and normal antenna cover materials. Another unknown factor for the determination of WAA is the information whether both, one or none of the antennas of a CML is wetted during a rain event.

To correct for WAA, several parametric correction schemes have been developed in the past. For the present data set, we
compared two of the schemes available from literature.

Schleiss et al. (2013) measured the magnitude and dynamics of WAA with one CML in Switzerland and derived a time-dependent WAA model. In this model, WAA increases at the beginning of a rain event to a defined maximum in a defined amount of time. From the end of the rain event on, WAA decreases again, as the wetted antenna dries. We ran this scheme with the proposed 2.3 dB of maximal WAA and a value of 15 for $\tau$, which determines the increase rate with time.

Leijnse et al. (2008) proposed a physical approach where the WAA depends on the microwave frequency, the antenna cover properties (thickness and refractive index) and the rain rate. A homogeneous water film is assumed on the antenna, with its thickness having a power law dependence on the rain rate. Higher rain rates cause a thicker water film and hence higher WAA. A factor $\gamma$ scales the thickness of the water film on the cover and a factor $\delta$ determines the non-linearity of the relation between rain rate and water film thickness. We adjusted the thickness of the antenna cover to 4.1 mm which we measured from an
antenna provided by Ericsson. We further adjusted $\gamma$ to 1.47E-5 and $\delta$ to 0.36 in such a way, that the increase of WAA with rain rates is less steep for small rain rates compared to the originally proposed parameters. The original set of parameters suppressed small rain events too much, because the WAA compensation attributed all attenuation to WAA. For strong rain events (>10 mm/h), the maximum WAA that is reached with our set of parameters is in the same range as the 2.3 dB used as maximum in the approach of Schleiss et al. (2013).





### 3.2.4 Derivation of rain rates

The estimation technique of rainfall from the WAA-corrected attenuation is based on the well known relation between specific path attenuation $k$ in dB/km and rain rate $R$ in mm/h

$$k = aR^b \tag{4}$$

with $a$ and $b$ being constants depending the on the frequency and polarization of the microwave radiation (Atlas and Ulbrich, 1977). In the currently most commonly used CML frequency range between 15 GHz and 40 GHz, the constants only show a low dependence on the rain drop size distribution. Using the $k$-$R$ relation, rain rates can be derived from the path integrated attenuation measurements that CMLs provide as shown in Fig. 3 d). We use values for $a$ and $b$ according to (ITU-R, 2005) which show good agreement with calculations from disdrometer data in southern Germany (Chwala and Kunstmann, 2019,

Fig. 3).

### 4   Results and Discussion

#### 4.1   Comparison of rain event detection schemes

The separation of wet and dry periods has a crucial impact on the accuracy of the rainfall estimation. We compare an approach from Schleiss and Berne (2010) to three modifications on their success in classifying wet and dry events as explained in 3.2.2.

The climatologic approach by Schleiss and Berne (2010) worked well for CMLs with moderate noise and when the fraction of times with rainfall over the analyzed periods corresponds to the climatological value. The median MDE was 0.33 and the median MCC of 0.43. The distribution of MDE and MCC values from all CMLs of this climatologic threshold were compared with the performance of two extensions, displayed in Fig. 4.

When we optimized the threshold for each CML for May 2018 and then applied these thresholds for the whole period, the

performance increased with a median MDE of 0.32 and median MCC of 0.46. The better performance of MDE and MCC values highlights the importance of a specific threshold for each individual CML accounting for their individual notion to fluctuate. The range of MDE and MCC values is wider than with the climatologic threshold, though. The wider range of MDE and MCC values, however, indicates that there is also a need for adjusting the individual thresholds over the course of the year. The 80th quantile-based method has the lowest median MDE with 0.27 and highest median MCC with 0.47. Therefore it miss-

classifies the least wet and dry periods compared to the other methods.

The threshold-based on the 80th quantile is independent from climatology and depends on the individual notion of a CML to fluctuate. Although the factor used to scale the threshold was derived from comparison to the reference data set as described in 3.2.2, it was stable over all seasons and for CMLs in different regions of Germany. Validating the scaling factor with other CML data sets could be a promising method for data scarce regions, as no external information is needed.

For single months, the MDE was below 0.20 as shown in Tab. 2, which still leaves room for an improvement of this rain event detection method. Enhancements could be achieved by adding information of nearby CMLs, if available. Also data from





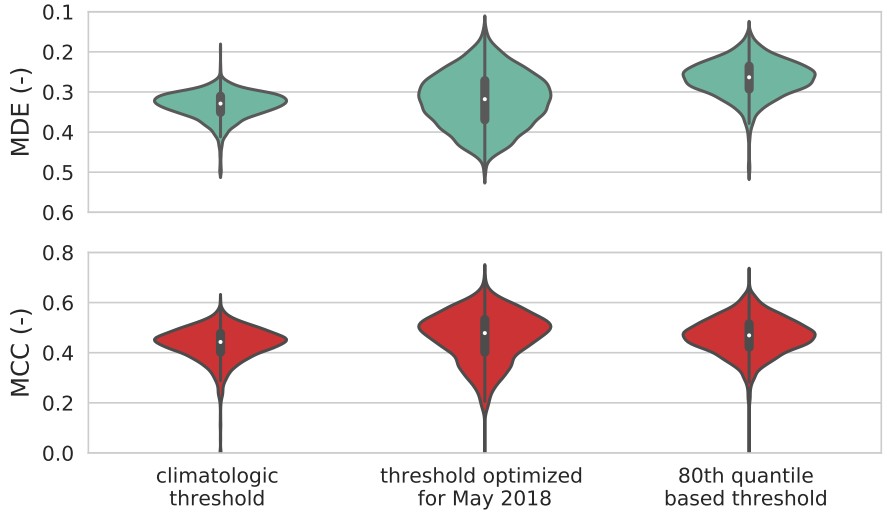

**Figure 4.** Mean detection error (MDE) and Matthews correlation coefficient (MCC) for three rain event detection schemes for the whole analysis period.

geostationary satellite could be used. Schip et al. (2017) found improvements of the rain event detection when using rainfall information from Meteosat Second Generation (MSG) satellite, which carries the Spinning Enhanced Visible and InfraRed Imager (SEVIRI) instrument.

All further processing, presented in the next sections, uses the method based on the 80th quantile.

**4.2   Performance of wet antenna attenuation schemes**

Two WAA schemes are tested and adopted for the present CML data set. Both are compared to uncorrected CML data and the reference in Fig. 5. Without a correction scheme, the CML-derived rainfall overestimated the reference rainfall by a factor of two when considering mean hourly rain rates, as displayed in Fig. 5a).

The correction by Schleiss and Berne (2010) produced comparable mean hourly rain rates with regard to the reference data
set. Despite its apparent usefulness to compensate for WAA, this scheme worked well only for stronger rain events. The mean detection error is higher than for the uncorrected data set, because small rain events are suppressed completely throughout the year. The discrepancy can also be a result of the link length of 7.6 km in our data set which is four times the length of the CML Schleiss et al. (2013) used. This might have an impact, since shorter CMLs have a higher likeliness that both antennas get wet. Furthermore, the type of antenna and antenna cover impacts the wetting during rain, as discussed in section 3.2.3.

With the method of Leijnse et al. (2008) the overestimation of the rain rate was also compensated well. It incorporates physical antenna characteristics and, what is more important, depends on the rain rate. The higher the rain rate, the higher the WAA compensation. This leads to less suppression of small events. The MDE is close to the uncorrected data sets and the PCC is also is higher, as displayed in 5b) and c). Therefore, this scheme is used for the evaluation of the CML-derived rain rates in the

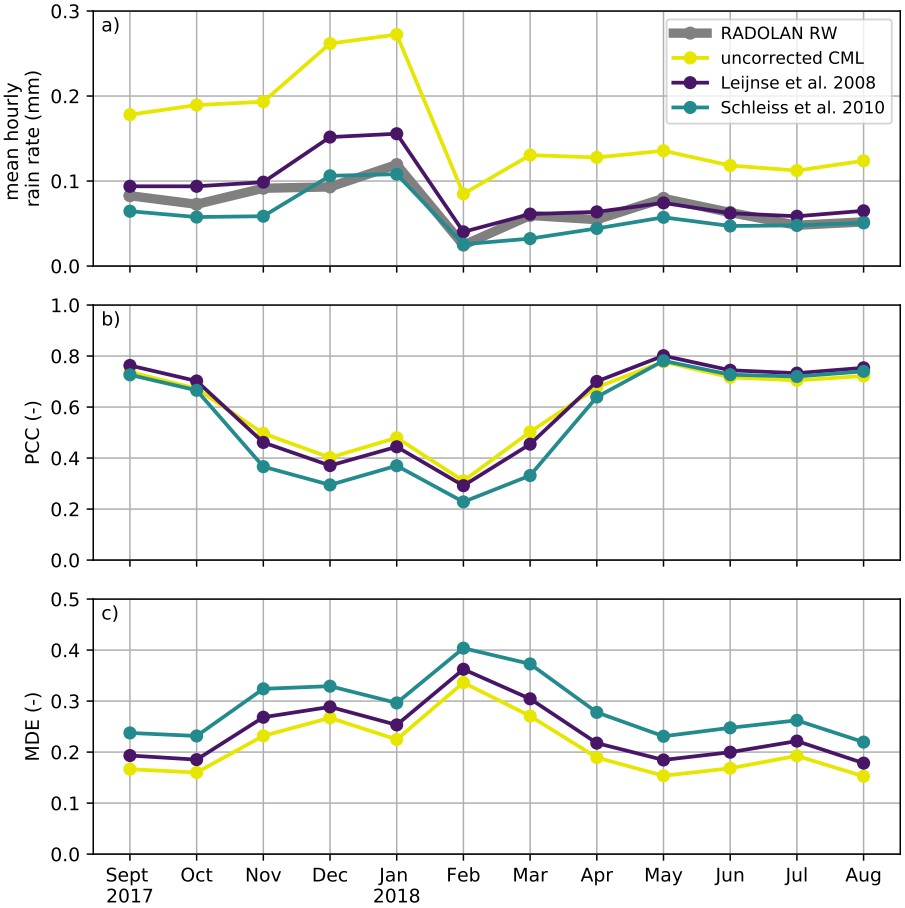

**Figure 5.** WAA compensation schemes compared on their influence on the a) mean hourly rain rate, b) the correlation between the derived rain rates and the reference and c) the mean detection error between the derived rain rates and the reference.

following section.

Both methods are parameterized, neglecting known and unknown interactions between WAA and external factors like temperature, humidity, radiation and wind. Current research aims to close this knowledge gap, but the feasibility for large scale networks like the one presented in this study is going to be a challenge as only TSL and RSL are available. A possible solution
5  is a WAA model based on the reflectivity of the antenna proposed by Moroder et al. (2019), which would have to be measured by future CML hardware. Another approach could be extending the analysis with meteorological model reanalysis products to be able to better understand WAA behavior in relation to meteorologic parameters like wind, air temperature, humidity and solar radiation.





## 4.3 Evaluation of CML derived rainfall

Rainfall information obtained from almost 4000 CMLs is evaluated against a reference data set, RADOLAN-RW. Hourly rain rates along the CML paths are used to generate scatter density plots shown in Fig. 6 and to calculate performance measures shown in Tab. 2.

When divided into seasons in Fig. 6, an occurrence of CML overestimation in the winter season (DJF) becomes apparent. This can be attributed to precipitation events with melting snow, occurring mainly from November to March. Melting snow can potentially cause as much as four times higher attenuation than a comparable amount of liquid precipitation (Paulson and Al-Mreri, 2011). Snow and ice on the covers of the antennas can also cause additional attenuation. This decrease of performance is also reflected in Tab. 2, where, on a monthly basis, the lowest values for PCC and highest for CV, MAE, RMSE and MDE

were found for DJF.

For the other seasons, CML rainfall and the reference have good correspondence on an hourly basis. In spring (MAM) and fall (SON), overestimation by CML rainfall is still visible but less frequent. This can be explained by the fact that, in the Central German Upland and the Alps, snowfall can occur from October to April. Best agreement between CML-derived rainfall and RADOLAN-RW is found for summer (JJA) months. September 2017 and May 2018 perform best when looking at the monthly

results, with higher PCC and lower CV values. Most likely, this is related to higher rain rates in those two month compared to the summer months JJA, which were exceptionally dry over Central Europe in 2018. The higher rain rates in September 2017 and May 2018 simplify the detection of rain events in the TRSL time series, and hence increase the overall performance. When compared over the whole analysis period, CML rainfall showed a notable overestimation for rain rates below 5 mm/h compared to the reference (not shown) due to the presence of non-liquid precipitation, but further showed a good agreement

for rain rates above 5 mm/h.

The rainfall sums of all CMLs are compared against the reference rainfall sums for each season in Fig 7. An overestimation of the CML derived rainfall sums can again be attributed to the presence of non-liquid precipitation and to the overestimation of hourly rain rates shown in Fig. 6. This overestimation is larger for higher rainfall sums. This could stem from more extensive snowfall in the mountainous parts of Germany which are also the areas with highest precipitation year round. Rainfall sums

close to zero can be the result from the quality control that we apply. The periods which are removed from CML time series are consequently not counted in the reference rainfall data set. Therefore, the rainfall sums in Fig. 6 are not representative for the rainfall sum over Germany for the shown period. The PCC for the four seasons shown in Fig. 7 range from 0.67 in DJF to 0.84 in JJA.

## 4.4 Rainfall maps

Interpolated rainfall maps of CML-derived rainfall compared to RADOLAN-RW are shown in Fig. 8 and Fig. 9. The maps have been derived using inverse distance weighting, representing each CML's rainfall value as one synthetic point at the center of the CML path. Interpolation is limited to regions which are at maximum 30 kilometers from the next CML away.




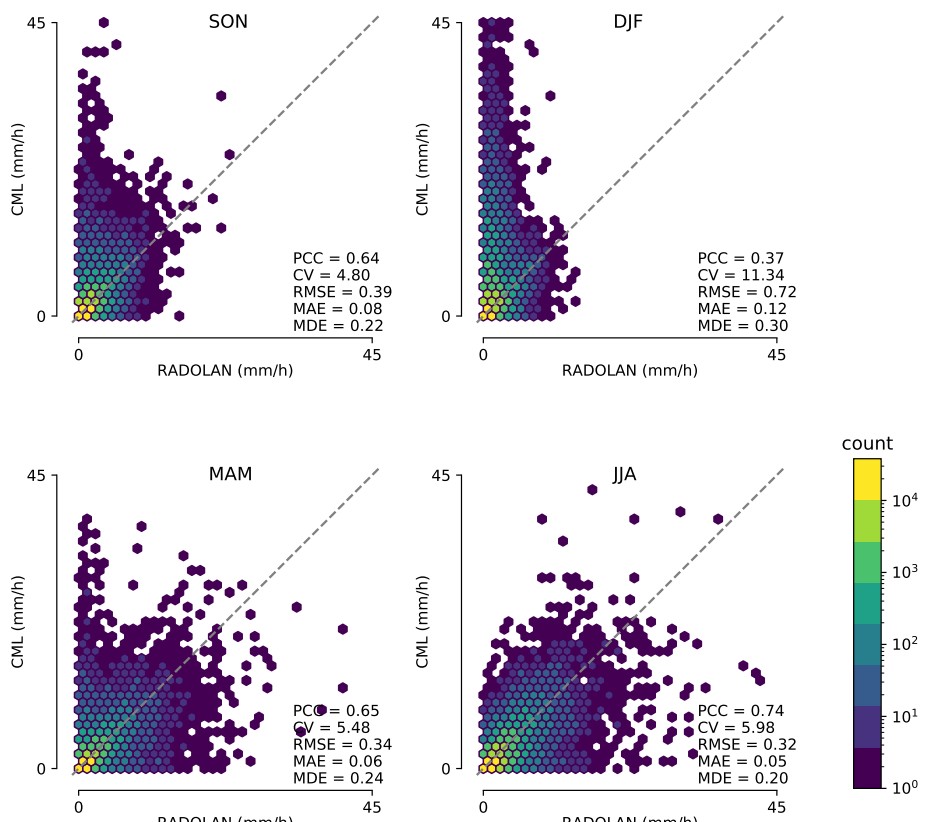

**Figure 6.** Seasonal scatter density plots between hourly CML-derived rainfall and RADOLAN-RW as reference.

**Table 2.** Performance measures between hourly CML-derived rainfall and RADOLAN-RW as reference.

| | mean | 2017 | | | | 2018 | | | | | | | |
| | | Sept | Oct | Nov | Dec | Jan | Feb | Mar | Apr | May | Jun | Jul | Aug |
|---|---|---|---|---|---|---|---|---|---|---|---|---|---|
| **PCC** | **.58** | .76 | .70 | .46 | .37 | .44 | .29 | .45 | .70 | .80 | .74 | .73 | .75 |
| **CV** | **6.64** | 3.90 | 4.38 | 5.78 | 9.20 | 6.80 | 16.08 | 6.39 | 5.36 | 4.29 | 5.58 | 6.45 | 5.65 |
| **MAE** | **.08** | .07 | .08 | .11 | .16 | .16 | .05 | .07 | .05 | .06 | .06 | .05 | .05 |
| **RMSE** | **.46** | .32 | .32 | .53 | .94 | .81 | .40 | .38 | .29 | .34 | .35 | .31 | .29 |
| **MDE** | **.25** | .19 | .18 | .26 | .28 | .25 | .36 | .30 | .21 | .18 | .20 | .22 | .17 |

Figure 8 shows a case of a 48 hour rainfall sum. The general distribution of CML-derived rainfall reproduces the pattern of the reference very well. Individual features of the RADOLAN-RW rainfall field are, however, missed due to the limited coverage by CMLs in certain regions.

Monthly CML derived rainfall fields also resemble the general patterns of RADOLAN-RW rainfall fields, as shown in Fig. 9.

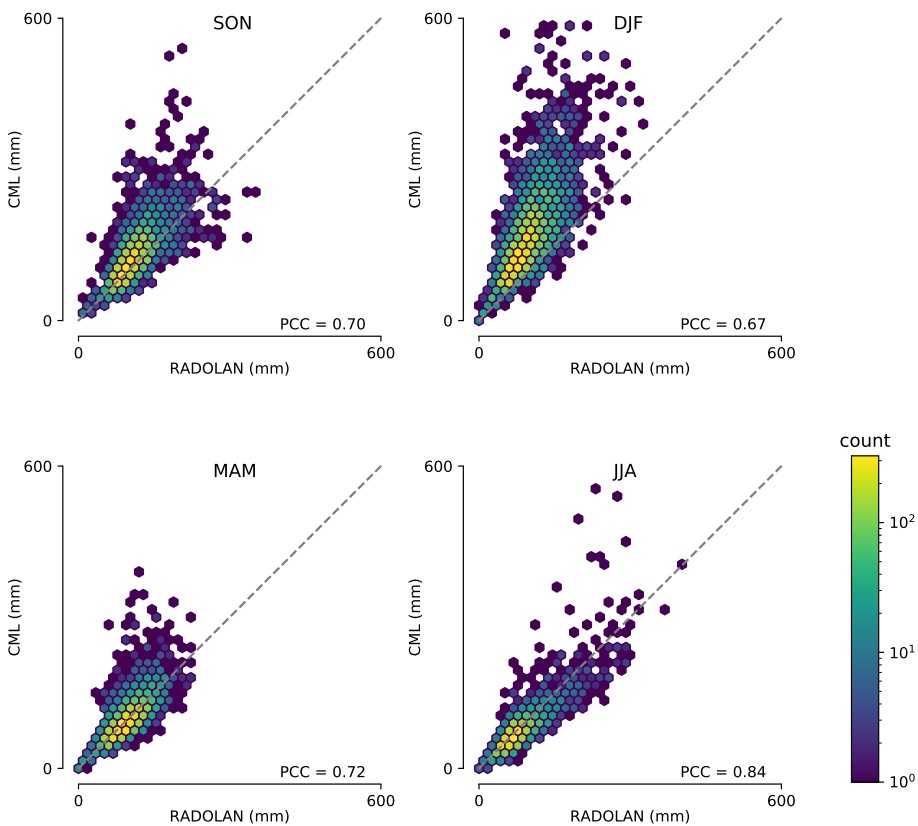

**Figure 7.** Seasonal scatter density plot of rainfall sums for each CMLs location derived from CML data and RADOLAN-RW as reference.

Summer months show a better agreement than winter months. This is a direct result of the decreased performance of CML-derived rain rates during the cold season, explained in section 4.3 and clearly visible in Fig. 7. Strong overestimation is also visible for a few individual CMLs, for which the filtering of erratic behavior was not always successful.

The derivation of spatial information from the estimated path-averaged rain rates could be improved by applying more sophis-

5  ticated techniques as described in 1. But, the errors in rain rate estimation for each CML contribute most to the uncertainty of CML-derived rainfall maps (Rios Gaona et al., 2015). Hence, within the scope of this work, we focus on improving the rainfall estimation at the individual CMLs. Therefore, we exclusively apply the simple inverse distance weighting interpolation and present the rainfall maps as an illustration of the potential of CMLs for countrywide rainfall estimation.

Taking into account that we compare to a reference data set derived from 17 C-band weather radars combined with more than

10  1000 rain gauges, the similarity with the CML-derived maps, which solely stem from the opportunistic usage of attenuation data, is remarkable.





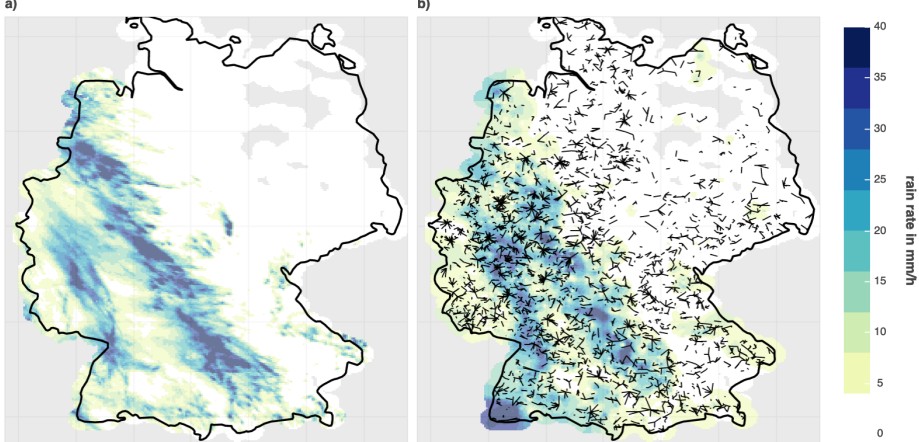

**Figure 8.** Accumulated rainfall for a 48 hour showcase from 12.05.2018 until 14.05.2018 for a) RADOLAN-RW and b) CML-derived rainfall. CML-derived rainfall is interpolated using a simple inverse distance weighting interpolation.

## 5  Conclusions

German wide rainfall estimates derived from CML data compared well with RADOLAN-RW, a hourly gridded gauge-adjusted radar product of the DWD. The methods used to process the CML data showed promising results over longer periods and several thousand CMLs across all landscapes in Germany, except for the winter season.

We presented the data processing of almost 4000 CMLs with a temporal resolution of one minute from September 2017 until August 2018. A CML data set of this size needs an automated processing workflow, which we developed. This workflow enabled us to test different processing methods over a large spatiotemporal scale.

A crucial processing step is the rain event detection from the TRSL, the raw attenuation data recorded for each CML. We use a scheme from (Schleiss and Berne, 2010) which uses the 60 minute rolling standard deviation RSD and a threshold. We derive

this threshold from a fixed multiple of the 80th quantile of the RSD distribution of each TRSL. Compared to the original, static threshold derived from a climatology, the 80th quantile reflects the general notion to fluctuate of each CML individually. We were able to reduce the amount of miss-classification of wet and dry events, reaching a yearly mean MDE of 0.27 with the summer months averaging below 0.20. Potential approaches for further decreasing the amount of miss-classifications could be the use of additional data sets. For example, cloud cover information from geostationary satellites could be employed to reduce

false wet classification, simply by defining periods without clouds as dry.

For the compensation of WAA, we compared and adjusted two approaches from literature. In order to evaluate WAA compensation approaches we used the reference data set. We were able to reduce the overestimation by WAA while maintaining the detection of small rain events, using an adjustment of the approach introduced by Leijnse et al. (2008). A WAA compensation without an evaluation with a reference data set is not feasible with the CML data set we use.

Compared to the reference data set RADOLAN-RW, the CML-derived rainfall compared well for periods with only liquid

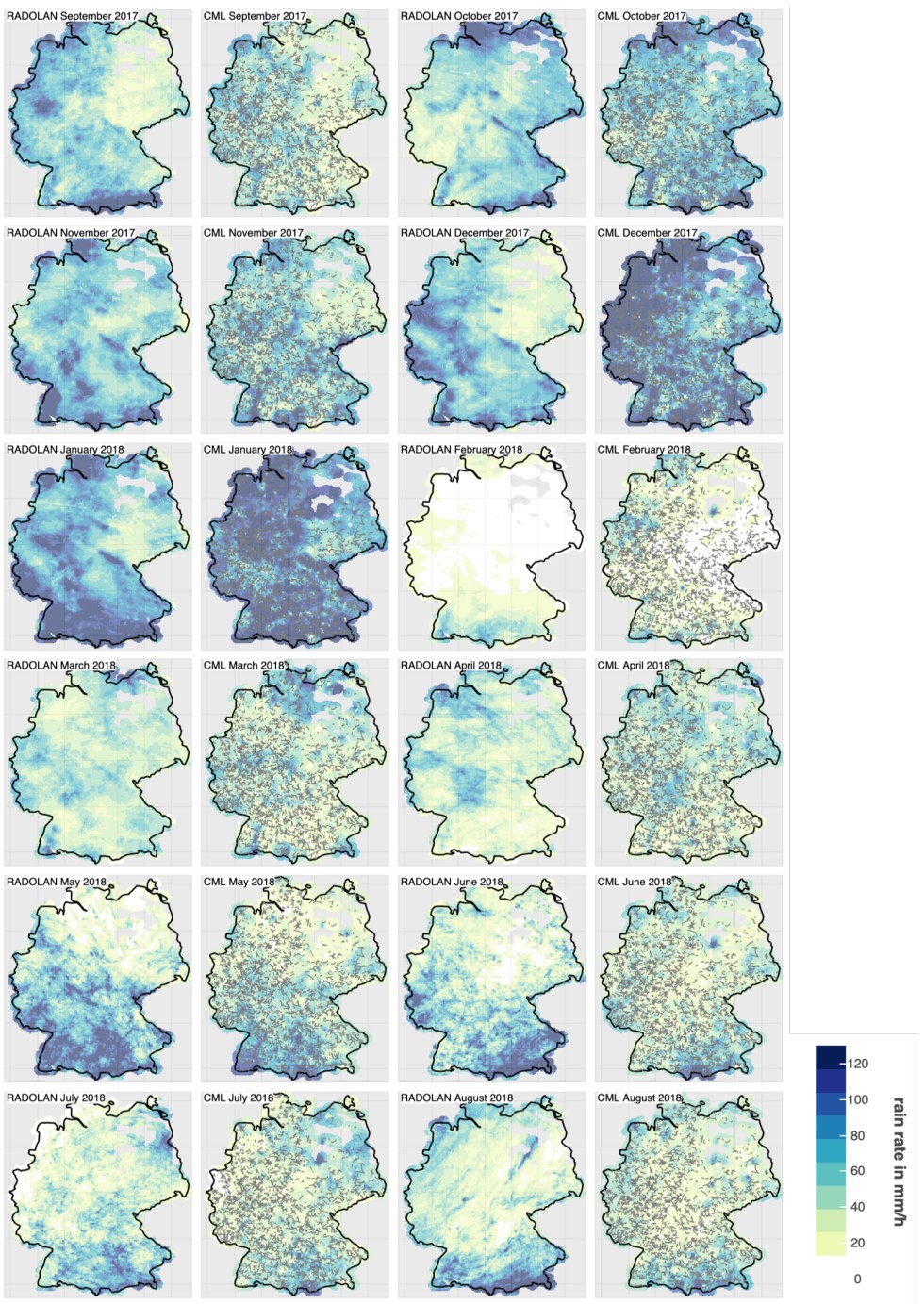

**Figure 9.** Monthly rainfall sums for RADOLAN-RW and CML derived rainfall from September 2017 until August 2018. CML derived rainfall is interpolated using a simple inverse distance weighting interpolation.



precipitation. For winter months, the performance of CML-derived rainfall is limited. Melting snow and snowy or icy antenna covers can cause additional attenuation resulting in overestimation of precipitation while dry snow cannot be measured with the used CMLs in our data set. We found high correlations for hourly, monthly and seasonal rainfall sums between CML-derived rainfall and the reference.

5  Qualitatively, we showed rainfall maps from RADOLAN-RW and CML-derived rainfall for a 48 hour showcase and all month of the analyzed period. A simple inverse distance weighting approach showed the plausibility of CML-derived rainfall maps. With the analysis presented in this study, the need for reference data sets in the processing routine of CML data is reduced, so that the opportunistic sensing of country-wide rainfall with CMLs is at a point, where it should be transferable to (reference) data scarce regions. Especially in Africa, where water availability and management are critical, this task should be challenged

10  as Doumounia et al. (2014) did already. But, CML derived rainfall can also complement other rainfall data sets in regions with high density of measurement networks and thus, substantially contribute to improved spatiotemporal estimations of rainfall.



*Code availability.* Code used for the processing of CML data can be found in the Python package pycomlink (pycomlink).

*Data availability.* CML data was provided by Ericsson, Germany and is not publicly available. RADOLAN-RW is publicly available through the Climate Data Center of the German Weather Service

*Author contributions.* MG, CC and HK designed the study layout and MG carried it out with contributions of CC and JP. Data was provided
5   by CC. Code was developed by MG with contributions of CC. MG prepared the manuscript with contributions from all co-authors.

*Competing interests.* The authors declare that they have no conflict of interest.

*Acknowledgements.* We thank Ericsson for the support and cooperation in the acquisition of the CML data. This work was funded by the Helmholtz Association of German Research Centres within the Project *Digital Earth*. We also like to thank the German Research Foundation for funding the projects *IMAP* and *RealPEP* and the Bundesministerium für Bildung und Forschung for funding the project *HoWa-innovativ*.



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
