# Peer review of "Rainfall estimation from a German-wide commercial microwave link network: Optimized processing and validation for one year of data"

_Hydrology and Earth System Sciences, 2019_

## Referee Comment (RC1) · Anonymous Referee #1 · 9 Sep 2019

General assessment.

The paper underpins the potential of rainfall estimation employing commercial microwave links (CMLs) from cellular telecommunication networks by using a full-year of data over entire Germany. The size of the dataset in terms of its coverage and number of CMLs is unprecedented. The original 1-minute temporal resolution is very high compared to other studies, which typically have 15-minute sampling strategies. Good results are obtained against a high-quality gauge-adjusted radar rainfall dataset, except for non-liquid precipitation, which was to be expected. Different rain event detection and wet antenna attenuation correction algorithms are compared. The evaluation

of CML-based path-averaged rainfall rates or sums and CML rainfall maps is fairly extensive. The paper is well written and clearly contributes to the upscaling of CMLs for rainfall monitoring. I congratulate the authors on obtaining such a large dataset, and the work they have done to facilitate this (Chwala et al., 2016).

Despite this positive assessment, I do have a number of more serious comments:

1) A comparison of the quality of CML-based rainfall estimates with those from other studies is completely missing. Please have a look at e.g. de Vos et al. (2019), who provide an overview for studies based on Dutch CML data, having a similar climate as many regions in Germany (see Table A1). Naturally, a fair comparison is only possible in case of similar thresholds and metrics, which may complicate some comparisons. It seems that no threshold is applied in your work, i.e. also zero rainfall estimates are incorporated. Please state this explicitly in your manuscript. You may also consider to show metrics for other thresholds, e.g. > 1mm. The performance can be highly dependent on the chosen threshold. This could facilitate the comparison with other studies. I miss the (relative) bias in the mean in e.g Figure 6 and Table 2.

2) It would be interesting to see scatter density plots or metrics of daily path-averaged rainfall (e.g. as. Fig. 6). It would also be interesting to see scatter density plots or metrics of hourly and daily interpolated rainfall. This would also help to compare results with those from other studies.

Specific comments.

1. pp. 1., l. 14-16: This is quite a bold statement. Though results are definitely good, correlation is not perfect and especially the coefficient of variation is rather high (Table 2). Although, part of this can be explained by representativeness errors, I think the statement is a bit too strong.

2. pp.2, l. 14-18: Add some information on geostationary satellite products. These have typically a fairly high temporal resolution of 15 min, but provide rather indirect

and therefore less accurate rainfall estimates. In addition, you could state that satellite products often have a limited spatial resolution, e.g. 0.1 degrees for GPM IMERG.

3. pp. 3., l. 21-22: Mention that all these gauges report hourly rainfall and add their spatial density (at least for the German ones), e.g. number of gauges per square kilometer.

4. pp. 3: Some more details on the reference dataset could be mentioned. What kind of rain gauge adjustment was performed (bias, spatial and what name)? Were dual-pol based algorithms employed, e.g. for clutter removal, attenuation correction, Kdp-R or Zdr-Zh-R rainfall retrieval? Why did you choose this radar rainfall product (perhaps: this is the shortest duration for which the radar product has been adjusted with gauges; even better radar products exist using more gauge data, but we wanted to show the performance with respect to a (near) real-time radar product). Is the used RADOLAN product really available in real time or is there a slight latency?

5. pp. 4, l. 5: Is this Ericsson network sufficient to provide full coverage over Germany, or is this one of the CML types used in the network of this company?

6. pp. 4, l. 11: How do you select the sub-link when two are available? Are there any criteria involved?

7. pp. 4, l. 19-23: Is the availability of radar data 100%? Please mention the availability. Is this availability taken into account, e.g. when comparing the radar-based versus CML-based rainfall maps?

8. pp. 6, l. 11: The authors could also add a reference to the overview paper by Messer et al. (2015).

9. pp. 6, l. 14: "requires repeatedly testing with the complete data set": Does this imply that part of the methodology has been optimized using the complete data set, i.e. that the evaluation is not entirely independent?

10. pp. 7: l. 8-11 & p.8, l. 1-2: Are these checks performed for each month? So that a

link may be discarded for one month, but be available for another month?

11. pp. 8, l. 23: Can you provide a reference for the 5 percent of the time it is raining? Here you assume that it is equally distributed over Germany.

12. pp. 8, l. 23-25: Could sources of error also constitute part of this 5 percent? So, assuming that 5 percent of the time it is indeed raining, this percentage would be too low if sources of error resulting in attenuation during dry periods have a similar magnitude.

13. pp. 8, l. 26-29 & pp. 9, l. 4-6: Can you provide somewhat more information on the optimization (e.g. which criteria)?

14. pp. 8, l. 33: Replace "for of" by "for".

15. pp. 9, l. 19: And what determines the decrease after an event?

16. pp. 9, l. 25: Do all these Ericsson antennas have the same cover?

17. pp. 9, l. 24: Is this 2.3 dB for one or two antennas? Is this value reasonable compared to the literature? In the wet antenna experiment from Van Leth et al. (2018) a value of 3-5 dB for one antenna was found (although this was not real rainfall).

18. pp. 10, l. 5: Replace "the on" by "on".

19. pp. 11, l. 18: Replace "also is" by "also".

20. pp. 13, l. 8 & pp. 18, l. 1: I expect that especially melting snow and ice on the covers gives rise to attenuation.

21. pp. 13, l. 20: I suppose that the reference is used to select rain rates above 5 mm/h?

22. pp. 14 & 15: For clarity I suggest to add that these are path-averages (i.e. not based on maps).

23. pp. 14, l. 14: You could add that e.g. in the southwestern part of Germany this is

the case.

24. pp. 15, l. 7: You could add that an advantage of the applied interpolation method is its robustness and speed.

25. pp. 18.: You could recommend that studying the quality of rainfall maps for shorter durations, e.g. 1 minute, would be an interesting follow-up study, especially for urban water management.

26. Figures 8 & 9: The tick marks do often not match the transition from one color scale to another.

27. pp. 16, l. 14-15: I think that algorithms using neighbouring CMLs are much more promising than satellite-based ones provided that the density of the CML network is high enough.

28. pp. 17: You could add as a recommendation to compare methods from different research groups on the same dataset, e.g. concerning rain event detection and wet antenna attenuation correction.

29. pp. 17, Figure 9: You could consider adding a map showing the relative or absolute difference of CML-based rainfall with respect to RADOLAN.

30. pp. 17. Are there any plans of merging CML data with RADOLAN? That could be an interesting recommendation. And what do you expect in terms of improved performance and especially for which areas (cities, valleys, ...)?

References.

Chwala, C., Keis, F., and Kunstmann, H.: Real-time data acquisition of commercial microwave link networks for hydrometeorological applications, Atmos. Meas. Tech., 9, 991–999, https://doi.org/10.5194/amt-9-991-2016, 2016.

Messer, H., & Sendik, O. (2015). A new approach to precipitation monitoring. IEEE Signal Processing Magazine, 32, 110– 122. https://doi.org/10.1109/MSP.2014.2309705

[Figure]

Van Leth, T. C., Overeem, A., Leijnse, H., and Uijlenhoet, R.: A measurement campaign to assess sources of error in microwave link rainfall 30 estimation, Atmospheric Measurement Techniques, 11, 4645–4669, 2018.

de Vos, L.W., A. Overeem, H. Leijnse, and R. Uijlenhoet, 2019: Rainfall Estimation Accuracy of a Nationwide Instantaneously Sampling Commercial Microwave Link Network: Error Dependency on Known Characteristics. J. Atmos. Oceanic Technol., 36, 1267–1283, https://doi.org/10.1175/JTECH-D-18-0197.1

---

## Referee Comment (RC2) · Anonymous Referee #2 · 3 Oct 2019

The authors present an analysis of rainfall estimation using minutely transmitted-minus-received signal level (TRSL) measurements from almost 4000 commercial microwave links (CMLs), located country-wide in Germany.

The fact that the authors have access to a very large database of minutely TSL/RSL measurements is unique, as, previous studies that presented a country-wide CMLs-based rainfall monitoring used a lower 15-minute sampling rate (and on top of that, some had access to the minimal and the maximal TSL/RSL values rather than the instantaneous values).

The presented rainfall estimation process follows the general steps established previously, including preparation of the data, baseline estimation, rain event detection, wet-antenna attenuation compensation, and rain-retrieval.

The authors compared the CML rain estimation outcome with the radar-based RADOLAN-RW data set, which shows, in general, good agreement.

Even though the presented study is very interesting, and can potentially contribute to this field of research, there are two main concerns that I feel the authors should address:

1. There are many different steps that the are being done in processing the data that include setting up different thresholds and margins (e.g., assuming that 5% of the time is classified as "rain", different moving-average window durations, different thresholds and percentile values from which the data is omitted, and so on). The problem here, is that there is no discussion regarding the logic behind selecting these specific parameters. It is very easy to "find the best parameters and thresholds" once you have a data-set used as ground-truth (in this case, the RADOLAN - which is later used for comparison). However, it is imperative to understand the actual process behind selecting these specific values, in order for the proposed methodology to be successfully deployed in different locations.

2. I find it lacking that no comparison with other established approaches of CML-based rain retrieval is being performed or discussed. Furthermore, the authors did not consider newer approaches for the different steps they perform (e.g., the wet-antenna or the baseline retrieval algorithms that are selected are based on algorithms published in 2008, 2010, and 2013, while there are many updated published newer studies. I am not saying that the decision to use the specific selected algorithms is incorrect, but, it should be explained why these specific algorithms are selected, with respect to other approaches that have been presented since.

---

## Referee Comment (RC3) · Anonymous Referee #3 · 24 Oct 2019

The authors present an interesting analysis of rain fall derived from an unique dataset of nearly 4000 CMLs measured at a 1-minute scale. The correspondence with RADOLAN-RW is in general good during summer and less so during winter. This is corresponds well with other studies and theory, but was able to this on a new larger scale than seen before. The study therefore shows the great potential of CMLs, especially in areas where there might be little other data sources available.

The paper is well constructed in general and will contribute to the further development of CML derived rain rates. There are a few points that I would like to see addressed however:

1. The reference dataset is based on gauge-adjusted hourly radar. While this offers the authors a source of data to compare link path derived rain rates with, it does not show the uniqueness of their dataset with a 1-minute resolution. The paper could for example benefit from an additional analysis of CMLs compared to rain gauge data with a high temporal resolution available at the DWD Climate Data Center. This analysis could be further extended by comparing hourly sums of rain gauge data with CML and RADOLAN derived rain rates (even though the RADOLAN data are of course adjusted using these same gauge data). While the rain gauges only offer point measurements, compared to the line measurements of the CMLs and the volumes of the radar it would give additional insight and offer the authors a chance to show the uniqueness of the dataset.

2. Like the first referee I think the paper might also benefit of analyzing the data at different thresholds, to show clearly how CMLs perform in at different rain intensities. It would also be good to clearly state how the filtering was performed. Is only a threshold applied to the RADOLAN data and how does this affect the CML data?

Finally a few minor comments:

P1, line 5: add a comma -> one year, spans

P2, line 11: this -> these

P2, line 12: remove the space before the .

P2, line 15: add a mention of the often limited spatial resolution of satellites

P4, line 13: The CML range is mentioned to be over 30km. In figure 2 there do not seem to be any CMLs beyond 30 km.

P6, fig2: The label on the x-axis should read (km) and not (m)

P9, line 24: Are all antennas of the same material and construction?

P13, line 34: What is the 30km range based on?

P15, line 5; But -> However

P18, line 10: But -> However

In general the text could benefit from added commas to improve readability.

—————————————————

---

## Author Response (AR1)

**Authors response to the Editor and the Referees for the paper:**

**Rainfall estimation from a German-wide commercial microwave link network: Optimized processing and validation for one year of data**

Dear Editor and Reviewers,

we appreciate your constructive comments which helped to improve our manuscript. This response letter is structured in the following way. We first summarize our general changes which follow the recommendations of the reviewers. Then we describe further general changes that we made in the analysis.

Regarding the recommendations of the reviewers, we implemented all changes in the way we proposed to do in our response to the reviewers, which we submitted to HESSD on 29[th] of November. Hence, the parts "Response to the comments of reviewer #N" below are exactly the same as the ones already submitted. Finally, we show a marked-up manuscript version of all changes.

In response to the recommended revision by the reviewers we introduced the following general changes to our manuscript:

1.  Besides the hourly and seasonal quantitative evaluation with scatter density plots and respective performance measures, we now also show this quantitative evaluation for daily aggregated rainfall sums for each season. (Updated Figure 6 and Section 4.3 Evaluation of CML derived rainfall)

2.  We introduced six subset criteria and rainfall thresholds for the comparison between path-averaged CML-derived and reference rain rates, to increase the comparability of the presented results and to evaluate the dataset under different aspects. (Extended Table 2 and updated text in Section 4.4 Performance measures for different subset criteria)

3.  With these subset criteria we were able to include a comparison with existing CML rainfall estimation studies. In addition, using the different subset criteria increases the comparability of our analysis with future studies. (New text in Section 4.4 Performance measures for different subset criteria)

4.  We now provide a quantitative evaluation of the CML-derived rainfall maps. We added a pixel-by-pixel comparison of the hourly CML rainfall maps with the reference RADOLAN-RW rainfall maps showing scatter density plots and performance measures on a monthly basis. We also added monthly link-based scatter density plots to show the difference between the map-based (pixe-by-pixel) and link-based (path-averaged along the CMLs) evaluation. (Extended Figure 8 and 9 and added new text in Section 4.5 Rainfall maps). We also decided to take the opportunity of this revision and try Kriging as a potential improvement over IDW. We tested several different approaches (details have been added to the end of section 4.5 Rainfall maps) for our complete data set. Performance metrics for the CML rainfall maps where equal or marginally better than the ones for IDW. Given that Kriging is computationally much more expensive than IDW and also sensitive to the selection of semivariogram parameters, we decided to keep IDW for the spatial interpolation of  CML rainfall information.

In addition to the revisions recommended by the reviewers, we changed the following parts in our analysis and the respective parts of the manuscript:

1. We increased the CML data availability by improving how we handle data from our data archive that stems from CMLs with a so called 1+1 hot-standby system. This led to almost 400 additional CMLs entering the processing after the removal of erratic behaving CMLs (Additional text in Section 2.2 Commercial microwave link data). As a consequence, individual results shown in the updated version of the manuscript differ slightly from the first version.

2. We now interpolate data gaps in raw TRSL time series up to five minutes. Furthermore we now also consider hours with at least 75% data availability in the analysis. We do this because short blackouts, which might stem from very high rainfall along a CML path, did lead to a complete neglection of the data of the affected CML in the respective hour in our old analysis. With this interpolation, we were able to increase the temporal data coverage by 0.5% (Additional text in Section 2.2 Commercial microwave link data). In consequence, individual results shown in the updated version of the manuscript differ slightly from the first version.

3. We implemented a dynamic coverage map around the CMLs which is now available for each time step and applied accordingly. We tested four different ranges from 10 to 50 km and used the 30 km range as a trade off between country-wide coverage and minimizing the uncertainty of the spatial interpolation. (Additional text in Section 4.5 Rainfall maps).

4. We corrected a mistake in the data processing for Fig. 7 which shows scatter density plots of seasonal rainfall sums from CMLs and the reference: In our old analysis we accidently removed CML-RADOLAN data pairs with false wet and missed wet classification before the seasonal aggregation. In the new scatter plots, shown in Fig, 6 i) - l), all available data pairs are included. The now correct plot shows a decreased performance of the CML rainfall estimation compared to the incorrect plot in the initial manuscript. This is due to the missclassifications (false wet, missed wet) that are now included. (Updated Figure 6 in Section 4.3 Evaluation of CML derived rainfall)

**Response to the comments of reviewer #1:**

We thank the reviewer for the valuable comments and the time to carefully examine the manuscript. In the following the comments of the reviewer are in black and our responses are in blue.

General assessment.

The paper underpins the potential of rainfall estimation employing commercial microwave links (CMLs) from cellular telecommunication networks by using a full-year of data over entire Germany. The size of the dataset in terms of its coverage and number of CMLs is unprecedented. The original 1-minute temporal resolution is very high compared to other studies, which typically have 15-minute sampling strategies. Good results are obtained against a high-quality gauge-adjusted radar rainfall dataset, except for non-liquid precipitation, which was to be expected. Different rain event detection and wet antenna attenuation correction algorithms are compared. The evaluation of CML-based path-averaged rainfall rates or sums and CML rainfall maps is fairly extensive. The paper is well written and clearly contributes to the upscaling of CMLs for rainfall monitoring. I congratulate the authors on obtaining such a large dataset, and the work they have done to facilitate this (Chwala et al., 2016).

Despite this positive assessment, I do have a number of more serious comments:

1) A comparison of the quality of CML-based rainfall estimates with those from other studies is completely missing. Please have a look at e.g. de Vos et al. (2019), who provide an overview for studies based on Dutch CML data, having a similar climate as many regions in Germany (see Table A1). Naturally, a fair comparison is only possible in case of similar thresholds and metrics, which may complicate some comparisons. It seems that no threshold is applied in your work, i.e. also zero rainfall estimates are incorporated. Please state this explicitly in your manuscript. You may also consider to show metrics for other thresholds, e.g. > 1mm. The performance can be highly dependent on the chosen threshold. This could facilitate the comparison with other studies. I miss the (relative) bias in the mean in e.g Figure 6 and Table 2.

Response: We agree with the reviewer that a comparison with other studies would add value to this study. We also agree that a fair comparison requires similar thresholds. Differences in the CML-network density and potential differences of reference datasets do not allow a purely quantitative comparison of the actual values, though. Nevertheless we see the benefit of making our results comparable with results of other studies.

We appreciate the comprehensive overview of CML-rainfall studies in the Netherlands given in de Vos et al. (2019) and will base our selection of comparisons on their Table A1.

Regarding the threshold, we can state that we applied a threshold of >= 0.1 mm/h for specific metrics, but did not made the use of this threshold clear enough throughout manuscript. The threshold only impacts the results of the performance metrics which are based on the differentiation between 'wet' and 'dry' periods. Also, for the density scatter plots we do not show pairs where both CML derived rain rates and the reference are dry. We will explain and highlight the use and implications of the threshold in the *Methods* and *Result* section.

Of course, the selection of another threshold can have an impact on the metrics, as low rain intensities are typically more frequent, but harder to detect with CMLs.

We suggest to make the following additions to our current analysis:
1. We will apply different thresholds: 0 mm, >= 0.1 mm and >= 1.0 mm for calculating the performance metrics of the path-averaged CML rain rates and discuss the results
2. We will discuss our results of path-averaged rain rates in comparison to the performances achieved in the respective studies from the Netherlands

3. As suggested by the reviewer, we will add the (relative) bias in the appropriate places e.g. Figure 6 and Table 2.

2) It would be interesting to see scatter density plots or metrics of daily path-averaged rainfall (e.g. as. Fig. 6). It would also be interesting to see scatter density plots or metrics of hourly and daily interpolated rainfall. This would also help to compare results with those from other studies.
Response: We agree that both suggestions are valuable for an increased comparability with other studies.

We suggest to make the following additions to our current analysis:
1. We will expand Fig. 6 to a 2x4 matrix, which will show hourly and daily path averaged scatter-density plots for each season and include metrics accordingly.
2. We will add a similar 2x4 matrix figure to the section *4.4 Rainfall Maps* which will present hourly and daily scatter-density plots derived from interpolated rainfall maps for each season and include metrics accordingly. For this figure and the calculation of the metrics we will use 'reference >= 0.1' mm as threshold as it is used in Overeem et al. (2016) and Rios Gaona et al. (2017).

Specific comments.
1. pp. 1., l. 14-16: This is quite a bold statement. Though results are definitely good, correlation is not perfect and especially the coefficient of variation is rather high (Table2). Although, part of this can be explained by representativeness errors, I think the statement is a bit too strong.
Response: We agree with the reviewer and will weaken the statement to the expression "good agreement".

2. pp.2, l. 14-18: Add some information on geostationary satellite products. These have typically a fairly high temporal resolution of 15 min, but provide rather indirect and therefore less accurate rainfall estimates. In addition, you could state that satellite products often have a limited spatial resolution, e.g. 0.1 degrees for GPM IMERG.
Response: We will extend the section about satellite products and will elaborate more on the spatial and temporal constraints and the differences between geostationary satellites and satellites in Low Earth orbits.

3. pp. 3., l. 21-22: Mention that all these gauges report hourly rainfall and add their spatial density (at least for the German ones), e.g. number of gauges per square kilometer.
Response: We will add more information on the rain gauges used to adjust the radar product which are automatic rain gauges of the German Weather Service with an hourly resolution and a spatial density of 0.003 gauges per square kilometer or one gauge per 325 square kilometers.

4. pp. 3: Some more details on the reference dataset could be mentioned. What kind of rain gauge adjustment was performed (bias, spatial and what name)? Were dual-pol based algorithms employed, e.g. for clutter removal, attenuation correction, Kdp-Ror Zdr-Zh-R rainfall retrieval? Why did you choose this radar rainfall product (perhaps:this is the shortest duration for which the radar product has been adjusted with gauges;even better radar products exist using more gauge data, but we wanted to show the performance with respect to a (near) real-time radar product). Is the used RADOLAN product really available in real time or is there a slight latency?
Response: We think we provided the necessary information on the reference data set as well as the literature providing further information. But we can add a very brief statement on the technology behind RADOLAN-RW (real-time, hourly, approx 15 minute delay, single-pol national radar composite adjusted with a mixture of additive and multiplicative rain gauge adjustment). Furthermore we will

extend the explanation on why we use this product and relate this explanation to the properties of the RADOLAN-RW.

5. pp. 4, l. 5: Is this Ericsson network sufficient to provide full coverage over Germany, or is this one of the CML types used in the network of this company?
Response: Indeed, we currently only have access to the CML network of one cellular provider and only to one CML type, the Ericsson MINI-LINK Traffic Node system. Therefore, we have limited spatial coverage in some regions of Germany, especially the north eastern part. Although, a 20 km buffer around the CMLs does not provide complete spatial coverage (in the north eastern part, as shown in Fig. 8), we have a high coverage over the rest of Germany.

6. pp. 4, l. 11: How do you select the sub-link when two are available? Are there any criteria involved?
Response: We always use the first listed sub-link of a given CML. We will add this information to the manuscript.

7. pp. 4, l. 19-23: Is the availability of radar data 100%? Please mention the availability. Is this availability taken into account, e.g. when comparing the radar-based versus CML-based rainfall maps?
Response: Yes, the availability of the processed reference, RADOLAN-RW along CML path, is 100%. When comparing path-integrated rainfall we exclude the pairs where CML derived rainfall is missing. For the quantitative comparison of radar and CML-based rainfall maps, which we will introduce in the revised version, we will use a coverage map that depends on CML data availability. This assures that we only validate parts of the grid which is within our defined coverage region, maximum 30km from a CML.

8. pp. 6, l. 11: The authors could also add a reference to the overview paper by Messer et al. (2015).
Response: We will add the reference by Messer et al. (2015).

9. pp. 6, l. 14: "requires repeatedly testing with the complete data set": Does this imply that part of the methodology has been optimized using the complete data set, i.e. that the evaluation is not entirely independent?
Response: We did not optimize individual steps of the method, except for the optimized threshold for May 2018 with the use of the MCC (p. 8, l. 24 ff.). During the development of the CML processing we did, however, also invent approaches that did not work well, when applied to the whole data set. Finding out that a new idea is a dead end, is sped up significantly with the parallelized workflow. We will rephrase the sentence to make this clearer.

10. pp. 7: l. 8-11 & p.8, l. 1-2: Are these checks performed for each month? So that a link may be discarded for one month, but be available for another month?
Response: These checks were performed for each month individually, yes. We will make this fact more clear in the manuscript.

11. pp. 8, l. 23: Can you provide a reference for the 5 percent of the time it is raining? Here you assume that it is equally distributed over Germany.
Response: We took the 5 percent from Schleiß and Berne (2010) who introduced the rolling standard deviation method. For the analyzed period of one year this value is more or less arbitrary because its reasoning is based on rainfall climatology. Of course, we are aware that this climatological threshold does not pay justice to the spatial and temporal variability of rainfall, but rather is a robust approach and simple way to provide a first rain event detection. In our study, this climatological approach, which

was suggested by the inventors of the method we use, serves as a reference method to show the improvement by the q80 method.

12. pp. 8, l. 23-25: Could sources of error also constitute part of this 5 percent? So, assuming that 5 percent of the time it is indeed raining, this percentage would be too low if sources of error resulting in attenuation during dry periods have a similar magnitude.

Response: Yes, this is possible and is yet another drawback of the climatological (5-percent) method. Our proposed q80-method should be a lot more robust against these errors because it takes the general noisiness of the TRSL as basis for setting the threshold. Still, this method is not free from errors in the form of misclassifications. But as can be seen in Figure 4, there is a clear improvement over the climatological method.

13. pp. 8, l. 26-29 & pp. 9, l. 4-6: Can you provide somewhat more information on the optimization (e.g. which criteria)?

Response: Of course, we will include this information in the respective parts of the manuscript. We will explain this in more detail by adding a text similar to e.g. "The optimal thresholds (pp. 7, l. 26ff.) are obtained in the following way: We process each individual CML in May 2018 with a range of possible thresholds for the rolling standard deviation method and calculate the binary measure MCC. We pick the threshold with the highest MCC for each individual CML and use it over the whole analysis period."

14. pp. 8, l. 33: Replace "for of" by "for".

Response: We will correct this typo.

15. pp. 9, l. 19: And what determines the decrease after an event?

Response: Schleiss and Berne (2013) found an exponential decrease after rain events in their data. But their WAA scheme does not explicitly model this decay since, as they write, "it is already contained in the values

of $a\_i$" ($a\_i$ is their total path attenuation after removal of the baseline). Hence, the WAA scheme with its exponential increase is only applied when the rain event detection considers the time step as wet, i.e. during a detected rain event.

16. pp. 9, l. 25: Do all these Ericsson antennas have the same cover?

Response: Based on the antennas that are used on Ericsson MINI-LINK Traffic Node CMLs that we have already had in our lab, we can say that there are at least two different types of covers. We took the value of 4.7 mm thickness from an 18 GHz antenna cover made from polycarbonate.

17. pp. 9, l. 24: Is this 2.3 dB for one or two antennas? Is this value reasonable compared to the literature? In the wet antenna experiment from Van Leth et al. (2018) a value of 3-5 dB for one antenna was found (although this was not real rainfall).

Response: 2.3 dB is the maximal WAA for the whole system, i.e. for both antennas. We took the value from Schleiss et al. (2013) who also used a CML for their study. Overeem et al. (2016) and the related studies with the Dutch CML data use a similar value. We, however, cannot say whether or not the real WAA does reach higher values at the CMLs in our data set. But we also do not know if both, one or none of the two antennas get wet during an individual rain event. Generally speaking, there is certainly room for improving WAA estimation methods. But in this study we want to apply a simple technique based on existing methods. We will explain our reasoning to choose the 2.3 dB better in the revised manuscript.

18. pp. 10, l. 5: Replace "the on" by "on".

Response: We will correct this typo.

19. pp. 11, l. 18: Replace "also is" by "also".
Response: We will correct this typo.

20. pp. 13, l. 8 & pp. 18, l. 1: I expect that especially melting snow and ice on the covers gives rise to attenuation.
Response: Indeed, we will add melting snow and ice on the covers to the causes of additional attenuation.

21. pp. 13, l. 20: I suppose that the reference is used to select rain rates above 5mm/h?
Response: Yes, the reference is used for selecting rain rates above 5mm/h. We will make this more clear in the revised manuscript.

22. pp. 14 & 15: For clarity I suggest to add that these are path-averages (i.e. not based on maps).
Response: We will add this in the caption of the figures.

23: pp. 14, l. 14: You could add that e.g. in the southwestern part of Germany this is the case.
Response: We will add this more precise description to the revised manuscript.

24. pp. 15, l. 7: You could add that an advantage of the applied interpolation method is its robustness and speed.
Response: We will add both advantages of IDW interpolation technique to the revised manuscript.

25. pp. 18.: You could recommend that studying the quality of rainfall maps for shorter durations, e.g. 1 minute, would be an interesting follow-up study, especially for urban water management.
Response: We will add this to the list of possible future possibilities of this methods and the presented data set.

26. Figures 8 & 9: The tick marks do often not match the transition from one color scale to another.
Response: We will place the tick marks on the correct position.

27. pp. 16, l. 14-15: I think that algorithms using neighbouring CMLs are much more promising than satellite-based ones provided that the density of the CML network is high enough.
Response: We agree that in dense CML networks these algorithms work well and will add this to the revised manuscript.

28. pp. 17: You could add as a recommendation to compare methods from different research groups on the same dataset, e.g. concerning rain event detection and wet antenna attenuation correction.
Response: Such a work would be an important step in making the work of different research groups really comparable. We will add this to the *Conclusion*.

29. pp. 17, Figure 9: You could consider adding a map showing the relative or absolute difference of CML-based rainfall with respect to RADOLAN.
Response: As we treat each CML only as a single point in the interpolation, a map showing the differences of CML and radar-based rainfall will show differences for almost all parts of the analyzed region. Also, since radar is a spatial sensor, the spatial variability in the radar reference is a lot higher than that of the interpolated CML rainfall fields, resulting in noisy difference plots. We will include scatter density plots of CML vs. radar-based rainfall maps as described in the response to the general comment 2), which will serve as basis of the comparison of both maps.

30. pp. 17. Are there any plans of merging CML data with RADOLAN? That could be an interesting recommendation. And what do you expect in terms of improved performance and especially for which areas (cities, valleys, ...)?

Response: Yes, at the moment we investigate merging CML data into the RADOLAN-process in a BMBF funded project called HoWa-innovativ (https://www.howa-innovativ.sachsen.de). As the reviewer guessed correctly, we expect the largest improvements in cities and valleys, where radar observations are hampered by ground clutter, beam-blockage or the vertical profile of reflectivity. We will add the potential of merging CML and radar data to the *Conclusion*.

References.

Chwala, C., Keis, F., and Kunstmann, H.: Real-time data acquisition of commercial microwave link networks for hydrometeorological applications, Atmos. Meas. Tech., 9,991–999, https://doi.org/10.5194/amt-9-991-2016, 2016.

Messer, H., & Sendik, O. (2015). A new approach to precipitation monitoring. IEEE Signal Processing Magazine, 32, 110– 122. https://doi.org/10.1109/MSP.2014.2309705

Overeem, A., Leijnse, H., & Uijlenhoet, R. (2016). Two and a half years of country-wide rainfall maps using radio links from commercial cellular telecommunication networks. *Water Resources Research*, *52*(10), 8039-8065.

Schleiss, M., & Berne, A. (2010). Identification of dry and rainy periods using telecommunication microwave links. *IEEE Geoscience and Remote Sensing Letters*, *7*(3), 611-615.

Schleiss, M., Rieckermann, J., & Berne, A. (2013). Quantification and modeling of wet-antenna attenuation for commercial microwave links. *IEEE Geoscience and Remote Sensing Letters*, *10*(5), 1195-1199.

Van Leth, T. C., Overeem, A., Leijnse, H., and Uijlenhoet, R.: A measurement campaign to assess sources of error in microwave link rainfall 30 estimation, Atmospheric Measurement Techniques, 11, 4645–4669, 2018.

de Vos, L.W., A. Overeem, H. Leijnse, and R. Uijlenhoet, 2019: Rainfall Estimation Accuracy of a Nationwide Instantaneously Sampling Commercial Microwave Link Network: Error Dependency on Known Characteristics. J. Atmos. Oceanic Technol., 36,1267–1283, https://doi.org/10.1175/JTECH-D-18-0197.1

**Response to the comments of reviewer #2:**

We thank the reviewer for the valuable comments and the time to carefully examine the manuscript. In the following the comments of the reviewer are in black and our responses are in blue.

The authors present an analysis of rainfall estimation using minutely transmitted-minus-received signal level (TRSL) measurements from almost 4000 commercial microwave links (CMLs), located country-wide in Germany.
The fact that the authors have access to a very large database of minutely TSL/RSL measurements is unique, as, previous studies that presented a country-wide CMLs-based rainfall monitoring used a lower 15-minute sampling rate (and on top of that,some had access to the minimal and the maximal TSL/RSL values rather than the instantaneous values).
The presented rainfall estimation process follows the general steps established previously, including preparation of the data, baseline estimation, rain event detection, wet-antenna attenuation compensation, and rain-retrieval.
The authors compared the CML rain estimation outcome with the radar-based RADOLAN-RW data set, which shows, in general, good agreement.

Even though the presented study is very interesting, and can potentially contribute to this field of research, there are two main concerns that I feel the authors should address:

1. There are many different steps that the are being done in processing the data that include setting up different thresholds and margins (e.g., assuming that 5% of the time is classified as "rain", different moving-average window durations, different thresholds and percentile values from which the data is omitted, and so on). The problem here,is that there is no discussion regarding the logic behind selecting these specific parameters. It is very easy to "find the best parameters and thresholds" once you have a data-set used as ground-truth (in this case, the RADOLAN - which is later used for comparison). However, it is imperative to understand the actual process behind selecting these specific values, in order for the proposed methodology to be successfully deployed in different locations.
Response: We agree that an understanding of all involved processes is very important and we will make the selection of parameters and thresholds clearer in the methodology section and discuss the decision where necessary. In detail we are going to address the following thresholds and parameters:

- p. 7, l. 11: length of filters to remove noisy data:
  We used two different filters to remove noisy/erratic CMLs for each month. For both filters we use thresholds which are in a range which is too far away from the average CML to make sense. We will explain the specific use of both filters in the revised manuscript more explicitly: With a five-hour moving standard deviation we filter CMLs which either have a very strong diurnal cycle or very noisy periods during a month. The other filter uses the moving window standard deviation approach by Schleiss et al. (2010) with a window length of one hour and a rain event detection threshold of 0.8. This threshold is conservative, meaning it selects only stronger rain events from a typical CML in our data set. When this threshold is exceeded more than one third of the time of a month, the respective CML is considered as too noisy and therefore is excluded from further analysis.
- p. 8, l. 22ff: 5 percent rainfall in Germany
  We took the 5 percent from Schleiß and Berne (2010), for the analyzed period of one year this value is more or less arbitrary because it is climatologic based. Of course, this climatological prerequisite cannot be fulfilled on either temporal or spatial scale but rather is a robust approach and simple way to provide a first rain event detection.
  (see also responses to comment 11 and 12 from Reviewer #1)

- p. 8, l. 33ff: 80th quantile and scaling factor
  The 80th quantile reflects the general amount of fluctuation of a CML without being influenced by the climatology as the threshold from Schleiß and Berne (2010) which explicitly uses the climatology. The scaling factor of q80 was calibrated for May 2018 with the optimal thresholds derived as explained at p. 8, l. 30ff. We also checked the derived scaling factor for other months and found them to be similar.
- p. 9, l. 25: γ and δ used in the WAA scheme from Leijnse et al. (2008)
  γ and δ are parameters of the logarithmic function of Leijnse et al. (2008)'s WAA model. As the WAA model from Schleiss et al. (2013) suppressed most of the small rain events, we chose γ and δ in a way that for small rain rates the WAA compensation is smaller, while high rain rates are treated with a correction in the same range as in the WAA model from Schleiss et al. (2013).

2. I find it lacking that no comparison with other established approaches of CML-based rain retrieval is being performed or discussed. Furthermore, the authors did not consider newer approaches for the different steps they perform (e.g., the wet-antenna or the baseline retrieval algorithms that are selected are based on algorithms published in 2008, 2010, and 2013, while there are many updated published newer studies. I am not saying that the decision to use the specific selected algorithms is incorrect, but, it should be explained why these specific algorithms are selected, with respect to other approaches that have been presented since.

Response: We will add a comparison to other CML studies as described in the answer to Reviewer 1. Also, we will explain the reasons which led to the selection of the processing steps in more detail in the manuscript e.g. for the WAA we selected a time (Schleiss et al. 2013) and a rain rate (Leijnse et al. 2008) dependent WAA model to compare both of them. The time dependent WAA model suppressed small rain rates too much, which is the reason we chose the rain rate dependent WAA model with certain parameters for γ and δ.

References:

Leijnse, H., Uijlenhoet, R., & Stricker, J. N. M. (2008). Microwave link rainfall estimation: Effects of link length and frequency, temporal sampling, power resolution, and wet antenna attenuation. *Advances in Water Resources*, *31*(11), 1481-1493.

Schleiss, M., & Berne, A. (2010). Identification of dry and rainy periods using telecommunication microwave links. *IEEE Geoscience and Remote Sensing Letters*, *7*(3), 611-615.

Schleiss, M., Rieckermann, J., & Berne, A. (2013). Quantification and modeling of wet-antenna attenuation for commercial microwave links. *IEEE Geoscience and Remote Sensing Letters*, *10*(5), 1195-1199.

**Response to the comments of reviewer #3:**

We thank the reviewer for the valuable comments and the time to carefully examine the manuscript. In the following the comments of the reviewer are in black and our responses are in blue.

The authors present an interesting analysis of rain fall derived from an unique dataset of nearly 4000 CMLs measured at a 1-minute scale.The correspondence with RADOLAN-RW is in general good during summer and less so during winter. This is corresponds well with other studies and theory, but was able to this on a new larger scale than seen before. The study therefore shows the great potential of CMLs, especially in areas where there might be little other data sources available.

The paper is well constructed in general and will contribute to the further development of CML derived rain rates. There are a few points that I would like to see addressed however:

1. The reference dataset is based on gauge-adjusted hourly radar. While this offers the authors a source of data to compare link path derived rain rates with, it does not show the uniqueness of their dataset with a 1-minute resolution. The paper could for example benefit from an additional analysis of CMLs compared to rain gauge data with a high temporal resolution available at the DWD Climate Data Center. This analysis could be further extended by comparing hourly sums of rain gauge data with CML and RADOLAN derived rain rates (even though the RADOLAN data are of course adjusted using these same gauge data). While the rain gauges only offer point measurements, compared to the line measurements of the CMLs and the volumes of the radar it would give additional insight and offer the authors a chance to show the uniqueness of the dataset.

Response: The goal of this study is to show the general performance of CML derived rainfall against a reference on a countrywide scale. We have chosen a spatial rainfall product (RADOLAN-RW) as reference in this study, because it allows us to validate the path-averaged rain rates of the CMLs. For a comparisons to rain gauges we would have to discuss and decide what a suitable maximal distance between CML-path and rain gauge is. Furthermore, with increasing path length, the path-averaged CML-derived rain rate estimates will be smoothed out compared to the point observations of the gauges. Both factors, the questions of distance between CML-path and rain gauge, as well as the effect of path-averaging will be more severe for short temporal aggregations. We have already done tests with the 1-minute rain gauge data from the DWD Climate Data Center and found that, even for the rain gauges in the vicinity of CMLs (gauge max. 2 km from CML-path, resulting in 191 CMLs with such a reference) we have to do temporal averaging to make rain rates comparable. We did not yet look on the effect of CML-path length, but it certainly will have an impact.

In conclusion, we agree that it would be interesting to compare the 1-minute CML-derived rainfall estimates to 1-minute rain gauge data. But since we can only do that in a meaningful way for a small subset of our data (191 CMLs have a gauge within 2km distance) and since it would introduce further uncertainties, we want to keep the existing analysis of this study homogeneous using only one reference dataset. Similarly, we believe it is beyond the scope of this study to compare rain gauge data with RADOLAN. We appreciate the reviewer's suggestion, though, and think it is worth working towards a separate study on the effect of spatio-temporal sampling differences between CML-, radar- and gauge data with high temporal resolution.

2. Like the first referee I think the paper might also benefit of analyzing the data at different thresholds, to show clearly how CMLs perform in at different rain intensities. It would also be good to clearly state how the filtering was performed. Is only a threshold applied to the RADOLAN data and how does this affect the CML data?

Response: We agree with the reviewer that the use of thresholds shows us the performance for different rain intensities. It further will increase the comparability to other studies. We will add this to our analysis as described in the response to Reviewer #1 general comment 1.

Finally a few minor comments:

P1, line 5: add a comma -> one year, spans
Response: We will add the comma.

P2, line 11: this -> these
Response: We will correct the typo.

P2, line 12: remove the space before the.
Response: We will correct the typo.

P2, line 15: add a mention of the often limited spatial resolution of satellites
Response: We will extend the section about satellite products and will elaborate more on the spatial and temporal constraints and the differences between geostationary and satellites in Lower Earth orbits.

P4, line 13: The CML range is mentioned to be over 30km. In figure 2 there do not seem to be any CMLs beyond 30 km.
Response: Indeed, this is a mistake, for the analyzed CML data set, there is no CML longer than 30 km. We will correct the manuscript accordingly.

P6, fig2: The label on the x-axis should read (km) and not (m)
Response: We will correct the typo.

P9, line 24: Are all antennas of the same material and construction?
Response: We currently have access to the CML network of one cellular provider and only to one CML type, the Ericsson MINI-LINK Traffic Node system. Based on the antennas from this network that we have already had in our lab, we can say that there are at least two different types of covers. We took the value of 4.7 mm thickness from an 18 GHz antenna cover made from polycarbonate.

P13, line 34: What is the 30km range based on?
Response: The 30 km range is a compromise. We have several larger gaps in the CML coverage and want to avoid that rainfall fields are generated too far away from the observations. We are aware that for extremely small scale convective events, the spatial decorrelation length for hourly rainfall will be smaller. But we also want to keep the spatial coverage of our CML rainfall field as high as possible. The value of 30 km was found to meet this requirement best. Results from van de Beek et al. (2012), who found the range of semi-variograms to be around 30 km for hourly rainfall in the summer months in the Netherlands, support our decision.

P15, line 5; But -> However
Response: We will correct the typo.

P18, line 10: But -> However
Response: We will correct the typo.

In general the text could benefit from added commas to improve readability.
Response: We will revise the manuscript under this perspective and try to increase the readability.

The available power resolution is 1 dB for TSL and 0.3 (with occasional jumps of 0.4 dB) for RSL. While the length of the CMLs ranges between a few hundred meters to  almost 30 km, most CMLs have a length of 5 to 10 km. They are operated with frequencies ranging from 10 to 40 GHz, depending on their length. Figure 2 shows the distributions of path lengths and frequencies. For shorter CMLs higher frequencies are used.

To derive rainfall from CMLs, we  used the difference between TSL and RSL, the transmitted minus received signal level

[Figure]

**Figure 1.** Map of the distribution of 3904 CMLs over Germany

(TRSL). An example of a TRSL time series is shown in Fig. 3a). To compare the rain rate derived from CMLs with the reference rain rate, we  resampled it from a minutely to an hourly resolution after the processing.

In our CML data set 2.2 percent are missing time steps due to outages of the data acquisition systems. Additionally 1.2 percent of the raw data show missing values (Nan) and 0.1 percent show default fill values (e.g. -99.9 or 255.0) of the CML hardware,

5 which we  excluded from the analysis.  In order to increase the data availability, we linearly interpolated gaps in raw TRSL time series which were up to five minutes long. This increased the data availability by 0.5 percent. On the one hand, these gaps can be the results of missing time steps and missing values but we also found cases where we suspect very high rainfall to be the

10 reason for short blackouts of a CML.

The size of  the complete CML data set is approximately 100 GB in memory. The data set is  continuously extended by the  operational data acquisition, allowing also the possibility of near-realtime rainfall estimation.

[Figure]

**Figure 2.** Scatterplot of the length against the microwave frequency of 3904 CMLs including the distribution of length and frequency.

**Table 1.** Adopted confusion matrix

| | | reference | |
|---|---|---|---|
| | | *wet* | *dry* |
| CML | *wet* | true wet (TP) | false wet (FP) |
| | *dry* | missed wet (FN) | true dry (TN) |

**3   Methods**

**3.1   Performance measures**

To evaluate the performance of the CML-derived rain rates against the reference data set, we used several measures which we calculated on an hourly basis. We defined a confusion matrix according to Tab. 1 where *wet* and *dry* refer to hours with and without rain, respectively.  The Matthew's correlation coefficient (MCC) summarizes the four values of the confusion matrix in a single measure (1) and is typically used as measure of binary classification in machine learning. This measure is accounting for the skewed ratio of wet and dry events. It is high only if the classifier is performing well on both classes.

$$\mathrm{MCC} = \frac{\mathrm{TP} * \mathrm{TN} - \mathrm{FP} * \mathrm{FN}}{\sqrt{(\mathrm{TP} + \mathrm{FP})(\mathrm{TP} + \mathrm{FN})(\mathrm{TN} + \mathrm{FP})(\mathrm{TN} + \mathrm{FN})}} \tag{1}$$

The mean detection error (MDE) (2) is introduced as a further binary measure focusing on the miss-classification of rain events.

$$\mathrm{MDE} = \frac{\frac{\mathrm{FN}}{\mathrm{n(wet)}} + \frac{\mathrm{FP}}{\mathrm{n(dry)}}}{2} \tag{2}$$

It is calculated as the average of missed wet and false wet rates of the contingency table from Tab. 1.

5    The linear correlation between CML-derived rainfall and the reference is expressed by the Pearson correlation coefficient (PCC). The coefficient of variation (CV) in (3) gives the distribution of CML rainfall around the reference expressed by the ratio of residual standard deviation and mean reference rainfall,

$$\mathrm{CV} = \frac{\mathrm{std} \sum (\mathrm{R_{CML}} - \mathrm{R_{reference}})}{\overline{\mathrm{R_{reference}}}} \tag{3}$$

where $\mathrm{R_{CML}}$ and $\mathrm{R_{reference}}$ are hourly rain rates of the respective data set. Furthermore, we computed the mean absolute error
10   (MAE) and the root mean squared error (RMSE) to measure the accuracy of the CML rainfall estimates. The relative bias is given as

$$\mathrm{bias} = \frac{\overline{(\mathrm{R_{CML}} - \mathrm{R_{reference}})}}{\overline{\mathrm{R_{reference}}}} \tag{4}$$

Often, in studies comparing CML derived rainfall and radar data, a threshold is used as a lower boundary for rainfall. The
15   performance measures, summarized in Tab. 2, were calculated with different subset criteria or thresholds. This gives insight on how CML derived rainfall compares to the reference for different rain rates and on how the large number of data points without rain influence the performance measures. Another reason for listing the performance measures with several thresholds is the increased comparability with other studies on CML rainfall estimation, which do not uniformly use the same threshold, see e.g. Table A1 in de Vos et al. (2019). Therefore, we defined a selection of subset criteria and thresholds and show performance
20   measures for data without any thresholds (*none*), for the data set with $\mathrm{R_{CML}}$ and $\mathrm{R_{reference}} < 0.1$ mm/h set to 0 mm/h, for two thresholds where at least $\mathrm{R_{CML}}$ or $\mathrm{R_{reference}}$ must be $> 0$ and $>= 0.1$ mm/h and two thresholds where $\mathrm{R_{reference}}$ must be $>= 0.1$ and $>= 1$ mm.

**3.2   From raw signal to rain rate**

As CMLs are an opportunistic sensing system rather than part of a dedicated measurement system, data processing has to be
25   done with care. Most of the CML research groups developed their own methods tailored to their needs and data sets. Overviews of these methods are summarized by Chwala and Kunstmann (2019), Messer and Sendik (2015) and Uijlenhoet et al. (2018). The size of our data set is a challenge itself. As TRSL can be attenuated by rain or other sources, described in Sect. 3.2.1 and only raw RSL TSL and RSL data is provided, the large size of the data set is of advantage but also a challenge. Developing and evaluating methods requires repeatedly testing with the complete data set. This requires was significantly sped up by the

[Figure]

**Figure 3.** Processing steps from the TRSL to rain rate. a) The TRSL is the difference of TSL - RSL, the raw transmitted and received signal level of a CML. b) The RSD (rolling standard deviation) of the TRSL with an exemplary threshold  shows the resulting wet and dry periods. c) The Attenuation is the difference between the baseline and the TRSL during wet periods. d) The derived rain rate is resampled to an hourly scale in order to compare it to the reference RADOLAN-RW

use of an automated processing workflow, which we implemented as a parallelized workflow on a HPC system using the Python packages *xarray* and *dask* for data processing and visual exploration. The major challenges which  arose from the processing of raw TRSL data into rain rates and the selected methods from literature are described in the following sections.

**3.2.1 Erratic behavior**

Rainfall is not the only source of attenuation of microwave radio along a CML path. Additional attenuation can be caused by atmospheric constituents like water vapor or oxygen, but also by refraction, reflection or multi-path propagation of the beam (Upton et al., 2005). In particular, refraction, reflection and multi-path propagation can lead to strong attenuation in the same magnitude as from rain. CMLs that exhibit such behavior have to be omitted due to their noisiness.

We excluded erratic CML data which was extremely noisy  or which showed drifts and jumps from our analysis  on a monthly basis. To deal with this erratic data, we applied the following sanity checks: We exclude individual CMLs if 1)  the five hour moving window standard deviation exceeds the threshold  2.0 for more then ten percent of a month,  which typically is the case for CMLs with either a strong diurnal cycle or very noisy periods during a month, or if 2) a one hour moving window standard deviation exceeds the threshold 0.8 more than 33 percent of the time in a month. This  filter is based on the approach for detecting rain events

in TRSL time series from Schleiss and Berne (2010), which we also use later on in our processing. For the filter, a fairly high threshold was used, which should only be exceeded for fluctuations stemming from real rain events. The reasoning of our filter is, that if the threshold is exceeded to often, here 33 percent of the time per month, the CML data shows an unreasonably high amount of strong  fluctuation. In total, the two sanity checks removed 1.1 percent from our CML data set. Together with the missing values that remain after interpolating data gaps of maximum five minutes in the TRSL time series, 4.2 percent of our data set are not available or not used for processing.

Jumps in data are mainly caused by single default values in the TSL which are described in Sect. 2.2. When we removed these default values, we are able to remove the jumps. TRSL can drift and fluctuate on daily and yearly scale (Chwala and Kunstmann, 2019). We could neglect the influence of these drifts in our analysis, because we dynamically derived a baseline for each rain event, as explained in  Sect. 3.2.2. We also excluded CMLs having a constant TRSL over a whole month.

**3.2.2 Rain event detection and baseline estimation**

The TRSL during dry periods can fluctuate over time due to ambient conditions as mentioned in the previous section. Rainfall produces additional attenuation on top of the dry fluctuation. In order to calculate the attenuation from rainfall, a baseline level of TRSL during each rain event has to be determined. We derived the baseline from the precedent dry period. During the rain event, this baseline was held constant, as no additional information on the evolution of the baseline level is available. The crucial step for deriving the baseline is to separate the TRSL time series into wet and dry periods, because only then the correct reference level before a rain event is used. By subtracting the baseline from TRSL, we derived the attenuation caused by rainfall which is shown in Fig. 3c).

The separation of wet and dry periods is essential, because the errors made in this step will impact the performance of rainfall estimation. Missing rain events will result in rainfall underestimation. False detection of rain events will lead to overestimation. The task of detecting rain events in the TRSL time series is simple for strong rain events, but challenging when the attenuation from rain is approaching the same order of magnitude as the fluctuation of TRSL data during dry conditions.

There are two essential concepts to detect rain events. One compares the TRSL of a certain CML to neighbouring CMLs (Overeem et al., 2016a) and the other investigates the time series of each CML separately (Chwala et al., 2012; Schleiss and Berne, 2010; Wang et al., 2012). We choose the latter one and  used a rolling standard deviation (RSD) with a centered moving window of 60 minutes length as a measure for the fluctuation of TRSL as proposed by Schleiss and Berne (2010).

It is assumed that RSD is high during wet periods and low during dry periods. Therefore, an adequate threshold  can be defined, which differentiates the RSD time series in wet and dry periods. An example of an RSD time series and a threshold is shown in Fig. 3b) where all data points with RSD values above the threshold are considered as wet.

Schleiss and Berne (2010) proposed the use of a RSD threshold derived from  rainfall climatology e.g. from nearby rain gauges. For our data set we  assumed that it is raining 5 percent of all minutes in Germany, as proposed by Schleiss and Berne (2010) for their CMLs in France. Therefore, we  used the 95 percent quantile of RSD as a threshold, assuming that the 5 percent of highest fluctuation of the TRSL time series refer to the 5 percent of rainy periods.

 We refer to this threshold as the climatologic threshold. We compared it to two new definitions of thresholds. We are aware that this threshold does not reflect the real climatology at each CMLs location, nevertheless this method is a rather robust and a simple approach which provides a first rain event detection.

For the first new definition, we derived the optimal threshold for each CML based on our reference data for the month of May 2018.  We used the same approach as for the climatologic threshold, but for each CML we tested a range of possible thresholds and calculated the binary measure MCC for each. For each CML we picked the threshold which produced the highest MCC in May 2018 and used it over the whole analysis period.

The second new definition to derive a threshold is based on the quantiles of the RSD, similarly to the climatologic threshold describe above. However, we propose to not focus on the fraction of rainy periods for finding the optimal threshold, since a rainfall climatology is likely not valid for individual years and not easily transferable to different locations. We took the 80th quantile of the RSD of each CML, which can be interpreted as a measure of the strength of the TRSL fluctuation during dry periods, and multiplied it by a constant factor to derive the individual threshold. The 80th quantile can be assumed to be more robust against missclassification than the climatologic threshold, because this quantile represents the general notion of each TRSL time series to fluctuate, rather than the percentage of time in which it is raining. We chose the 80th quantile, since it is very unlikely that it is raining more than 20 percent of the time in a month in Germany.

To find the right factor, we selected the month of May 2018 and fitted a linear regression between the optimal threshold for each CML and the 80th quantile. The optimal threshold was derived beforehand with a MCC optimization from the reference. We then used this factor for all other months in our analysis. Additional, we found it to be similar for all months of the analyzed period.

**3.2.3 Wet antenna attenuation**

Wet antenna attenuation is the attenuation caused by water on the cover of a CML antenna. With this additional attenuation, the derived rain rate overestimates the true rain rate (Schleiss et al., 2013; Zinevich et al., 2010). The estimation of WAA is complex, as it is influenced by partially unknown factors, e.g. the material of the antenna cover. van Leth et al. (2018) found differences in WAA magnitude and temporal dynamics due to different sizes and shapes of the water droplets on hydrophobic and normal antenna cover materials. Another unknown factor for the determination of WAA is the information whether both, one or none of the antennas of a CML is wetted during a rain event. To correct for WAA, several parametric correction schemes have been developed in the past. For the present data set, we compared two of the schemes available from literature.

Schleiss et al. (2013) measured the magnitude and dynamics of WAA with one CML in Switzerland and derived a time-dependent WAA model. In this model, WAA increases at the beginning of a rain event to a defined maximum in a defined amount of time. From the end of the rain event on, WAA decreases again, as the wetted antenna is drying off. We ran this scheme with the proposed 2.3 dB of maximal WAA for both antennas together. This is also similar

to the WAA correction value of 2.15 dB, which Overeem et al. (2016b) derived over a 12-day period in their data set. For $\tau$, which determines the increase rate with time , we chose 15 minutes. The decrease of WAA after a rain event is not explicitly modelled, because this WAA scheme is only applied for time steps, which are considered wet from the rain event detection, which has to be carried out in a previous step.

5   Leijnse et al. (2008) proposed a physical approach where the WAA depends on the microwave frequency, the antenna cover properties (thickness and refractive index) and the rain rate. A homogeneous water film is assumed on the antenna, with its thickness having a power law dependence on the rain rate. Higher rain rates cause a thicker water film and hence higher WAA. A factor $\gamma$ scales the thickness of the water film on the cover and a factor $\delta$ determines the non-linearity of the relation between rain rate and water film thickness. We adjusted the thickness of the antenna cover to 4.1 mm which we measured from an

10  one antenna provided by Ericsson. We are aware of the fact, that antenna covers have different thicknesses. But since we do not have this information for the actual antennas that are used by the CMLs of our data, we use this values, as the best one available. 
[revised manuscript text omitted]

rain rates, daily rainfall sums and seasonal sums of each CML with the respective performance measures. Furthermore, scatter density plots  of hourly, path-averaged rain rates and rain rates from interpolated rainfall maps are compared for each month in Fig.  8 and Fig. 9. Looking at the differences between the seasons in 6, it is evident, that CMLs are prone to produce significant rainfall overestimation during the cold season (DJF). This can be attributed to precipitation events with melting snow, occurring mainly from November to March. Melting snow can potentially cause as much as four times higher attenuation than a comparable amount of liquid precipitation (Paulson and Al-Mreri, 2011). Snow, ice and their melt water on the covers of the antennas can also cause additional attenuation.  A decrease of the seasonal performance measures also reflects this effect, as the lowest values for PCC and highest for CV, MAE, RMSE, BIAS and MDE are found for DJF.

[Figure]

**Figure 6.** Seasonal scatter density plots of CML-derived rainfall and path-averaged RADOLAN-RW data for hourly, a) - d), daily, e) - h) and seasonal, i) - l) aggregations with respective performance metrics calculated from all available data pairs.

seasons, CML rainfall and the referencehave good correspondence on an hourly basis. The largest overestimation occurs at low rain rates of the reference. At higher reference rain rates, which most likely are those stemming from liquid precipitation, there is far less overestimation. In spring (MAM) and fall (SON), overestimation by CML rainfall is still visible, but less frequent. This can be explained by the factthat,, that in the Central German Upland and the Alps, snowfall can occur from October to April. Best agreement between CML-derived rainfall and RADOLAN-RW is found for summer (JJA) months. September 2017 and May 2018 perform best when looking at the monthly results, with higher PCC and lower CV values. Most likely, this is related to higher rain rates in those two month compared to the summer months JJA, which were exceptionally dry over Central Europe in 2018. The higher rain rates in September 2017 and May 2018 simplify the detection of rain events in the TRSL time series, and hence increase the overall performance. When compared over the whole analysis period, CML rainfall showed a notable overestimation for rain rates below 5 mm/h compared to the reference (not shown)

The temporal aggregation to daily rainfall sums and the respective performance measures are shown in 6e)-h). The general relation between CML derived rainfall and the reference is similar on both the hourly and daily scale. The BIAS is identical

for the daily aggregation. The RMSE and MAE are higher due to the  higher rain sums. The overestimation during the winter month is unchanged. The accumulated rainfall sums of  individual CMLs are compared against the reference rainfall  accumulation for each season in Fig  6i) - l). The overestimation of the CML derived rainfall sums in DJF, and partly SON and MAM, can again be attributed to the presence of non-liquid precipitation. This overestimation is larger for higher rainfall sums. This could  be the result of more extensive snowfall in the mountainous parts of Germany, which are also the areas with highest precipitation year round. Rainfall sums close to zero can be the result from the quality control that we  have applied. Periods with missing data in CML time series are consequently not counted in the reference rainfall data set. Therefore, the rainfall sums in Fig. 6 are not representative for the rainfall sum over Germany for the shown period. The PCC for the four seasons shown in Fig.   6i)-l) range from 0.42 in MAM to 0.57 in JJA.

**4.4   Performance measures for different subset criteria**

Tab. 2 gives an overview of monthly performance measures for different subsets of CML-derived and path-averaged reference rainfall. In the following, we will discuss the effects of the different subset criteria and then compare our results to previous CML rainfall estimation studies.

For all subset criteria, best performance measures are found during late spring, summer and early fall. Highest PCC values are reached when all data pairs, including true dry events, are used to calculate the measures. When very light rain (< 0.1 mm/h) is set to zero on an hourly basis, the performance measures stay very similar, with the exception of CV and BIAS, which show a slight increase in performance. This means that, even when very small rain rates < 0.1 mm are produced, they do not change rainfall sums too much.

When either $R_{CML}$ or $R_{reference}$ have to exceed 0 mm/h, the performance measures are worse than with all data, because all 0 mm/h pairs are removed. When the same subset criteria is set to 0.1 mm/h, a good agreement in the range of very small rain rates below 0.1 mm/h between both data becomes apparent, because the performance measure get worse without them.

To examine the performance of the CML derived rainfall during rain events detected in the reference, two thresholds are selected, where the reference must be above 0.1 and 1 mm/h, respectively. With this thresholds, all false wet classifications are removed before the calculation of the measures. The PCC with this thresholds is still high for the non-winter months. The CV is reduced, while MAE and RMSE are higher due to higher mean rain rates. The biggest differences can be observed in the bias, where the influence of false wet detection and the overestimation of CMLs over 0.1 and 1 mm/h reduce the bias.

When discussing these measures in relation to previous studies on CML rainfall estimation, the selection of the thresholds is of great importance due to their strong impact on the performance measures. de Vos et al. (2019) showed a collection of CML-studies in Table A1. Their own data set consists of 1451 CMLs for the summer of 2016 in the Netherlands and is at the same time the most comparable to the data set presented here, considering size, temporal aggregation and selected threshold.

**Table 2.**  Monthly performance measures between path averaged, hourly CML-derived rainfall and RADOLAN-RW as reference for subset criteria and thresholds.

| | subset criteria (mm) |  mean | 2017 Sept | Oct | Nov | Dec |  Jan | 2018 Feb | Mar | Apr |
|---|---|---|---|---|---|---|---|---|---|---|
| **PCC (-)** | none |  **0.62** | 0.78 | 0.73 | 0.46 | 0.36 | 0.43 | 0.27 | 0.45 | 0.74 |
| | light rain to 0 |  **0.62** | 0.78 | 0.73 | 0.46 | 0.36 | 0.43 | 0.27 | 0.45 | 0.74 |
| | cml or ref > 0 |  **0.58** | 0.74 | 0.68 | 0.38 | 0.28 | 0.35 | 0.20 | 0.37 | 0.71 |
| | cml or ref >= 0.1 |  **0.54** | 0.70 | 0.64 | 0.34 | 0.23 | 0.31 | 0.13 | 0.32 | 0.68 |
| | ref >= 0.1 |  **0.58** | 0.73 | 0.71 | 0.38 | 0.28 | 0.35 | 0.22 | 0.39 | 0.73 |
| | ref >= 1 | **0.51** | 0.65 | 0.64 | 0.32 | 0.17 | 0.27 | 0.12 | 027 | 0.67 |
| **CV (-)** | none | **7.01** | 3.80 | 4.40 | 6.09 | 11.4 | 7.62 | 18.5 | 6.82 | 5.20 |
| | light rain to 0 | **7.19** | 3.88 | 4.51 | 6.23 | 11.64 | 7.75 | 18.28 | 7.06 | 5.33 |
| | cml or ref > 0 | **3.03** | 1.73 | 2.00 | 2.96 | 5.59 | 3.85 | 6.82 | 3.09 | 2.19 |
| | cml or ref >= 0.1 | **2.42** | 1.40 | 1.64 | 2.51 | 4.78 | 3.35 | 5.19 | 2.53 | 1.67 |
| | ref >= 0.1 | **1.69** | 1.05 | 1.06 | 1.92 | 3.61 | 2.67 | 3.25 | 1.90 | 1.11 |
| | ref >= 1 | **1.11** | 0.73 | 0.69 | 1.24 | 2.27 | 1.73 | 2.18 | 1.14 | 0.70 |
| **MAE (mm/h)** | none | **0.08** | 0.08 | 0.08 | 0.11 | 0.17 | 0.17 | 0.05 | 0.07 | 0.05 |
| | light rain to 0 | **0.08** | 0.08 | 0.07 | 0.11 | 0.17 | 0.16 | 0.05 | 0.07 | 0.05 |
| | cml or ref > 0 | **0.41** | 0.38 | 0.36 | 0.46 | 0.71 | 0.64 | 0.37 | 0.35 | 0.30 |
| | cml or ref >= 0.1 | **0.64** | 0.58 | 0.53 | 0.64 | 0.97 | 0.86 | 0.66 | 0.53 | 0.49 |
| | ref >= 0.1 | **0.72** | 0.64 | 0.57 | 0.70 | 1.02 | 0.91 | 0.68 | 0.55 | 0.54 |
| | ref >= 1 | **1.40** | 1.16 | 1.05 | 1.40 | 2.02 | 1.73 | 1.73 | 1.25 | 1.09 |
| **RMSE (mm/h)** | none | **0.48** | 0.34 | 0.33 | 0.56 | 1.08 | 0.94 | 0.46 | 0.41 | 0.29 |
| | light rain to 0 | **0.48** | 0.35 | 0.33 | 0.56 | 1.08 | 0.94 | 0.46 | 0.41 | 0.29 |
| | cml or ref > 0 | **1.06** | 0.75 | 0.71 | 1.16 | 2.18 | 1.84 | 1.25 | 0.90 | 0.68 |
| | cml or ref >= 0.1 | **1.34** | 0.94 | 0.87 | 1.38 | 2.58 | 2.14 | 1.70 | 1.12 | 0.90 |
| | ref >= 0.1 | **1.45** | 1.01 | 0.90 | 1.47 | 2.66 | 2.22 | 1.68 | 1.15 | 0.96 |
| | ref >= 1 | **2.33** | 1.59 | 1.43 | 2.36 | 4.02 | 3.33 | 3.48 | 1.97 | 1.61 |
| **BIAS (%)** | none | **30** | 20 | 34 | 11 | 79 | 39 | 67 | 7 | 21 |
| | light rain to 0 | **29** | 20 | 34 | 11 | 80 | 40 | 67 | 7 | 20 |
| | cml or ref > 0 | **30** | 20 | 34 | 11 | 79 | 39 | 67 | 7 | 21 |
| | cml or ref >= 0.1 | **29** | 20 | 33 | 11 | 80 | 40 | 67 | 7 | 20 |
| | ref >= 0.1 | **-4** | -1 | -1 | -15 | 36 | 14 | -6 | -20 | -10 |
| | ref >= 1 | **-9** | -4 | -9 | -24 | 22 | 2 | -16 | -21 | -12 |
| **MDE** | none | **0.23** | 0.20 | 0.19 | 0.24 | 0.27 | 0.23 | 0.35 | 0.29 | 0.22 |

Their performance measures are a r$^2$ of 0.27, a CV of 3.43 and a BIAS of 23 % at an hourly resolution with the threshold *cml or ref >0*. While in the presented data set the squared PCC (0.30) is higher and the CV is lower, it has a higher bias than the data of de Vos et al. (2019). Even with many similarities there still are differences e.g. in the sampling strategy, which inhibit a true quantitative comparison of the measures. Another similar study was carried out by Rios Gaona et al. (2015), who gave

5 a r$^2$ of 0.36, a CV of 1.2 and a BIAS of -14.3 % for 1514 CMLs in the Netherlands for 12 selected rainy (summer) days. The measures are in the same range, especially when only considering summer month for the used threshold of *cml or ref >= 0.1* from Tab. 2. Again, the sampling strategy, the explicit selection of twelve days and the differences in the CML network and the radar product, limit the comparability. In order to get true comparability, one should either use the same data set to compare different processing approaches or the same processing approach to compare different data sets.

10 ## 4.5 Rainfall maps

Interpolated rainfall maps of CML-derived rainfall compared to RADOLAN-RW are shown in Fig. 7, Fig. 8 and Fig.  9. The respective CML maps have been derived using inverse distance weighting  (IDW) with the RADOLAN-RW grid as target grid and on an hourly basis. Each CML rainfall value is represented as one synthetic point observation at the center of the  CMLs

15 path. For each pixel of the interpolated rainfall field the nearest 12 synthetic CML observation points are taken into account. Weights decrease with the distance $d$ in km, according to $d^{-2}$. After the interpolation, we masked out grid cells further away than 30  km from a CML path, for each individual time step. Hence, hourly rainfall maps derived from CMLs are only produced for areas with data coverage. We applied the same mask to the reference data set on an hourly basis to increase the comparability between both data sets. For the aggregated rainfall maps,

20 we summed up the interpolated, individually masked, hourly rainfall fields. As an example, Fig. 7 shows 48  hours of accumulated rainfall in May 2018. The general distribution of CML-derived rainfall reproduces the pattern of the reference very well and the rainfall sums of both data sets are similar. Individual features of the RADOLAN-RW rainfall field are, however, missed due to the limited coverage by CMLs in certain regions.

 A qualitative comparison of monthly aggregation of the hourly rainfall maps is shown

25 in Fig. 8 and Fig. 9. The CML-derived rainfall fields resemble the general patterns of the RADOLAN-RW rainfall fields. Summer months show a better agreement than winter months. This is a direct result of the decreased performance of CML-derived rain rates during the  winter season, explained in  Sect. 4.3. Strong overestimation is also visible year round for a few individual CMLs, for which the filtering of erratic behavior was not  successful.

30 A quantitative comparison of the CML-derived rainfall maps to the reference is shown in the third column of Fig. 8 and Fig. 9. For these scatter density plots we used all hourly pixel values of the respective month within the 30 km coverage mask. During the winter month, CMLs show strong overestimation. This is a direct result of non-liquid precipitation as described in Sect. 4.3. From May to August 2018 the reference shows very high rain intensities between 50 and 100 mm/h, which are not produced by the CML rainfall maps. This can be attributed to several reasons. First, CML-derived rainfall, which serves as basis for the

[Figure]

**Figure 7.** Accumulated rainfall for a 48 hour showcase from 12.05.2018 until 14.05.2018 for a) RADOLAN-RW and b) CML-derived rainfall. CML-derived rainfall is interpolated using a simple inverse distance weighting interpolation. A coverage mask is 30 km around CMLs is used.

interpolation, is path-averaged, with a typical path length in the range of 3-15 km. This means, that the rainfall estimation of a single CML represents an average of several RADOLAN-RW grid cells which smoothes out the extremes. Second, due to the interpolation, rainfall maxima in the CML rainfall maps can only occur at the synthetic observation points at the center of each CML. Third, rainfall is only observed along the CMLs path and even with almost 4000 CMLs across Germany, the spatial
5   variation of rainfall cannot be fully resolved. In particular in summer, small convective rainfall events might not intersect with CML paths and hence cannot appear in the CML-derived IDW interpolated rainfall fields.

Considering this, the effect of different coverage ranges around the CMLs has to be taken into account. For the map based comparison in Fig. 8 and Fig. 9 we tested several distances from 10 to 50 km. For the presented results we choose 30 km as a trade off between minimizing the uncertainty of the spatial interpolation and the goal to reach country wide coverage with the
10   produced rainfall maps. van de Beek et al. (2012) found an averaged range of around 30 km for summery semi-variograms of 30 years of hourly rain gauge data in the Netherlands, which can be used to justify/enforce our choice. With a 10 km coverage range, the performance measures are better than the ones for 30 km, which are shown in Fig. 8 and Fig. 9. Monthly PCC values show an increase of around 0.05 and the bias is reduced by 3 to 5 percent. Nevertheless, with a coverage of 10 km around the CMLs, coverage gaps emerge not only in the north-eastern part of Germany, but also in the south eastern part. Vice versa, with
15   a 50 km coverage range, the country wide coverage is almost given, while the performance measures are worse compared to 30 km (PCC shows a decrease between 0.03 and 0.05). Overall, the difference of the performance measures of the 10 and 50 km coverage mask is limited in most parts of Germany by the high density of CMLs, which already lead to an almost full coverage with the 10 km mask.

In order to highlight the differences between a map-based and link-based comparison Fig. 8 and Fig. 9 also show hourly
20   link-based scatter density plots for each month. The differences in the performances measures for the warm months support the

[Figure]

**Figure 8.** Monthly aggregations of hourly rainfall maps from CMLs compared to RADOLAN-RW from September 2017 until February 2018. For each month two scatter density plots are shown, one for pixel-by-pixel comparison of the hourly maps (map-based comparison), and one for the comparison of the path-averaged rainfall along the individual CMLs (link-based comparison).

[Figure]

**Figure 9.** Monthly aggregations of hourly rainfall maps from CMLs compared to RADOLAN-RW from March until August 2018. For each month two scatter density plots are shown, one for pixel-by-pixel comparison of the hourly maps (map-based comparison), and one for the comparison of the path-averaged rainfall along the individual CMLs (link-based comparison).

qualitative impression, that the map-based comparison perform worse. The interpolation is prone to introduce an underestimation for areas which are more distant to the CML observations. During the winter months, this underestimation compensates the overestimation of the individual CMLs which is due to wet snow and ice covered antennas. Hence, because the two errors compensate each other by chance, this results in slightly better map-based performance measures compared to the link-based

5 measures for the winter months. Nevertheless, rainfall estimation with CMLs for months with non-liquid precipitation is considerably worse than for summer months in all spatial and temporal aggregations.

The derivation of spatial information from the estimated path-averaged rain rates could be improved by applying more sophisticated techniques as described in  Sect. 1. We have already carried out several experiments using Kriging, to test one of these potential improvements over IDW. We followed the approach of Overeem et al. (2016b) and adjusted

10 the semivariogram parameters on a monthly basis based on the values from van de Beek et al. (2012). We also tried fixed semivariogram parameters and parameters estimated from the individual CML rainfall estimates for each hour. In conclusion, we, however, only found marginal or no improvements of the performance metrics of the CML rainfall maps. Combined with the drawback of Kriging that the required computation time is significantly increased (approximately 10 to 100 times slower than IDW, depending e.g. on the number of neighboring points used by a moving krigging window), we thus decided to keep

15 using the simple, yet robust and fast IDW interpolation. Furthermore, it is important to note that the errors in rain rate estimation for each CML contribute most to the uncertainty of CML-derived rainfall maps (Rios Gaona et al., 2015). Hence, within the scope of this work, we  focused on improving the rainfall estimation at the individual CMLs.

20 Taking into account that we compare to a reference data set derived from 17 C-band weather radars combined with more than 1000 rain gauges, the similarity with the CML-derived maps, which solely stem from the opportunistic usage of attenuation data, is remarkable.

25

**5   Conclusions**

German wide rainfall estimates derived from CML data compared well with RADOLAN-RW, a hourly gridded gauge-adjusted radar product of the DWD. The methods used to process the CML data showed promising results over  one year

30 and several thousand CMLs across all landscapes in Germany, except for the winter season.

We presented the data processing of almost 4000 CMLs with a temporal resolution of one minute from September 2017 until August 2018.  We developed a parallelized processing work flow, which could handle the size of this large data set. This workflow enabled us to test and

compare different processing methods over a large spatiotemporal scale.

A crucial processing step is the rain event detection from the TRSL, the raw attenuation data recorded for each CML. We  used a scheme from (Schleiss and Berne, 2010) which uses the 60 minute rolling standard deviation RSD and a threshold. We  derived this threshold from a fixed multiple of the 80th quantile of the RSD distribution of each TRSL. Compared to the original  threshold using the 95th quantile, which is based on rainfall climatology, the 80th quantile reflects the general notion  of each CML's TRSL to fluctuate. We were able to reduce the amount of miss-classification of wet and dry events, reaching a yearly mean MDE of 0.27, with an average of the MDE for summer months below 0.20. Potential approaches for further decreasing the amount of miss-classifications could be the use of additional data sets. For example, cloud cover information from geostationary satellites could be employed to reduce false wet classification, by, as a first simple approach, defining periods without clouds as dry. Another opportunity would be, to additionally implement algorithms exploiting information of neighboring CMLs.

For the compensation of WAA, the attenuation caused by water droplets on the cover of CML antennas, we compared and adjusted two approaches from literature. In order to evaluate WAA compensation approaches, we used the reference data set. We were able to reduce the overestimation  caused by WAA, while maintaining the detection of small rain events, using an adjustment of the approach introduced by Leijnse et al. (2008).  The compensation of WAA without an evaluation with a reference data set is not feasible with the CML data set we use.

Compared to the reference data set RADOLAN-RW, the CML-derived rainfall  performs well for periods with only liquid precipitation. For winter months, the performance of CML-derived rainfall is limited. Melting snow and snowy or icy antenna covers can cause additional attenuation resulting in overestimation of precipitation, while dry snow cannot be measured  at the frequencies and the TRSL quantizations the CMLs in our data set . We found high correlations for hourly, monthly and seasonal rainfall sums between CML-derived rainfall and the reference. To increase the comparability of our analysis with existing and future studies on CML rainfall estimation we calculated all performance metrics for different subset criteria, e.g. requiring that either CML or reference rainfall is larger than 0 mm. We found the performance measures of this study to be in accordance with similar CML studies, although the comparability is limited due to differences of the CML and reference data sets. CML-derived rainfall maps calculated with a simple, yet robust inverse distance weighting  interpolation showed the plausibility of CMLs as an stand-alone rainfall measurement system.

With the analysis presented in this study, the need for reference data sets in the processing routine of CML data is reduced, so that the opportunistic sensing of country-wide rainfall with CMLs is at a point, where it should be transferable to (reference) data scarce regions. Especially in Africa, where water availability and management are critical, this task should be challenged as Doumounia et al. (2014) did already. The high temporal resolution of the presented data set can be used in future studies, e.g. for urban water management. In addition, CML derived rainfall can also complement other rainfall data sets, e.g. to improve the radar data adjustment in RADOLAN in regions with high

 CML density and regions, like mountain ranges, where radar data is often compromised. Thus, CMLs can contribute substantially to improve the spatiotemporal estimations of rainfall.

[revised manuscript text omitted]